*Report*

# Bacterial RNA sensing by TLR8 requires RNase 6 processing and is inhibited by RNA 2'O-methylation

Ivanéia V Nunes [1,2,7], Luisa Breitenbach [1,2,7], Sarah Pawusch [3], Tatjana Eigenbrod [1,4], Swetha Ananth [5], Paulina Schad [1], Oliver T Fackler [5,6], Falk Butter [3], Alexander H Dalpke [1,2] & Lan-Sun Chen [1,2]

## Abstract

TLR8 senses single-stranded RNA (ssRNA) fragments, processed via cleavage by ribonuclease (RNase) T2 and RNase A family members. Processing by these RNases releases uridines and purine-terminated residues resulting in TLR8 activation. Monocytes show high expression of RNase 6, yet this RNase has not been analyzed for its physiological contribution to the recognition of bacterial RNA by TLR8. Here, we show a role for RNase 6 in TLR8 activation. BLaER1 cells, transdifferentiated into monocyte-like cells, as well as primary monocytes deficient for *RNASE6* show a dampened TLR8-dependent response upon stimulation with isolated bacterial RNA (bRNA) and also upon infection with live bacteria. Pretreatment of bacterial RNA with recombinant RNase 6 generates fragments that induce TLR8 stimulation in RNase 6 knockout cells. 2'O-RNA methyl modification, when introduced at the first uridine in the UA dinucleotide, impairs processing by RNase 6 and dampens TLR8 stimulation. In summary, our data show that RNase 6 processes bacterial RNA and generates uridine-terminated breakdown products that activate TLR8.

**Keywords** Innate Immunity; Toll-Like Receptor 8; RNase 6; Nucleic Acid Recognition; RNA Modifications
**Subject Categories** Microbiology, Virology & Host Pathogen Interaction; RNA Biology; Signal Transduction

## Introduction

TLR7 and TLR8 detect short single-stranded RNA (ssRNA) that can be derived from viral and bacterial RNA (Mann and Hornung, 2021). TLR8 is highly expressed in monocytes, neutrophils, and myeloid DCs and is required for bacterial ssRNA (bRNA) recognition from several bacteria, including *Streptococcus pyogenes*, *Staphylococcus aureus*, and *Enterococcus faecalis* ((Cervantes et al, 2011; Krueger et al, 2015; Bergstrøm et al, 2015; Eigenbrod et al, 2015). TLR8 also senses infections with intact, whole bacteria, thus arguing for a physiological contribution to the defense against infections (Ugolini et al, 2018; Eigenbrod and Dalpke, 2015; Hafner et al, 2019; Vierbuchen et al, 2017). Proper TLR8 activation induces a MyD88-mediated signaling cascade, culminating in the release of NF-κB-dependent cytokines and IRF-5-dependent type I IFN (Bergstrøm et al, 2015; Heinz et al, 2020).

TLR8 is a horseshoe-shaped homodimeric receptor, with each of the TLR8 protomers contributing with an apical and a concave binding site (Tanji et al, 2015). Small synthetic agonists, such as the imidazoquinolinone derivates R848 and CL097, can efficiently activate the receptor solely by binding to the apical site (Tanji et al, 2013). Mechanistic studies uncovered that although uridine ribomononucleosides can bind to the apex site as well, TLR8 activation still requires simultaneous binding at the concave site by 2- or 3-mer purine-2′3′-cyclophosphate-terminated oligoribonucleotides (ORNs) (Tanji et al, 2013, 2015). These findings show that TLR8 senses short fragments instead of long ssRNA, thus RNA processing is required.

RNase T2 and members of the RNase A family were recently identified as contributors to ssRNA processing: their breakdown fragments showed a strong immunostimulatory potential for TLR8 activation, and lack of RNase T2 or RNase 2 resulted in reduced activation by RNA (Greulich et al, 2019; Ostendorf et al, 2020). Regarding their cleavage selectivity, it was shown that RNase T2 preferentially cleaves between purines and uridine (Greulich et al, 2019), and RNase A family members, in general, cleave in between pyrimidines and purines (reviewed in (Boix et al, 2013)). ssRNA fragments sensed by TLR8 are products of a synergistic cleavage by these RNase families.

Among the several canonical RNase A family members, RNase 2 was the first one reported to be involved in the generation of ssRNA fragments in the endolysosome of THP-1 cells (Ostendorf et al, 2020). Also known as Eosinophil-derived Neurotoxin (EDN), RNase 2 is mostly expressed in eosinophils, yet showing also expression in monocytes to lesser extent according to the Human Protein Atlas (Rosenberg, 2015; Uhlen et al, 2019). Thus, although ssRNA processing by RNase 2 may result in uridine-terminated breakdown products activating TLR8, its role in monocytes remains unsure. However, monocytes express RNase 6, another RNase A

[1]Dept. of Infectious Diseases, Medical Microbiology and Hygiene, Medical Faculty, Heidelberg University, Heidelberg, Germany. [2]University Hospital Heidelberg, Heidelberg, Germany. [3]Institute of Molecular Virology and Cell Biology, Friedrich-Loeffler-Institute, Greifswald, Germany. [4]Institute of Laboratory Medicine, SLK Clinics Heilbronn GmbH, 74078 Heilbronn, Germany. [5]Department of Infectious Diseases, Center for Integrative Infectious Disease Research (CIID), Integrative Virology, Heidelberg University, Medical Faculty Heidelberg, Heidelberg, Germany. [6]German Center for Infection Research (DZIF), Heidelberg Partner Site, Heidelberg, Germany. [7]These authors contributed equally: Ivanéia V Nunes, Luisa Breitenbach. ✉E-mail: alexander.dalpke@med.uni-heidelberg.de

family member, to a higher extent (Ostendorf et al, 2020; Tong et al, 2023).

RNase 6 exhibits extracellular antimicrobial peptide (AMP) activity against a range of bacteria at low micromolar concentrations (Becknell et al, 2015; Pulido et al, 2016). RNase 6 is highly cationic which might foster the binding to negatively charged components of the bacterial membrane thus inducing bacterial cell death (Becknell et al, 2015; Rosenberg and Dyer, 1996; Tong et al, 2023). Expression of RNase 6 seems to be limited to immune cells such as monocytes, neutrophils, and dendritic cells (DCs) (Rosenberg and Dyer, 1996; Tong et al, 2023). It is detected at high concentrations in immune cell-rich tissues such as the spleen and thymus (Becknell et al, 2015). RNase 6 was shown to protect against urinary tract infections (Becknell et al, 2015) and *RNASE6* transgenic mice showed increased protection against infection with uropathogenic *Escherichia coli* (Ruiz-Rosado et al, 2023). Of note, RNase 6 expression overlaps with TLR8 in monocytes, neutrophils, and myeloid DCs, and its role in the processing of ssRNA was recently reported (Tong et al, 2023). Yet, whether and how the ribonuclease activity of RNase 6 may contribute to the defense of bacterial infections remains unclear.

TLR8 activation is often studied using synthetic oligoribonucleotides (ORN). However, a plethora of posttranscriptional structural RNA modifications are known to occur in pro- and eukaryotic RNA, and RNA modifications are contributing to avoid recognition of self-RNA (Karikó et al, 2005). Although prokaryotic RNA is not as heavily modified as eukaryotic RNA, the same features can serve as a strategy to escape from immune recognition (Eigenbrod and Dalpke, 2015; Galvanin et al, 2020). Previous work of our group showed that RNA 2′O-methylation (2′O-Me) within prokaryotic tRNA can affect recognition by endosomal TLRs (Gehrig et al, 2012). In a recent report, 2′O-Me-modified uridines also prevented RNase 6 cleavage (Tong et al, 2023) and, consequently, the generation of fragments for TLR8 activation.

We here sought to investigate RNase 6 with respect to its role in the processing of bacterial RNA (bRNA) upon infection. We show that bRNA from gram-positive bacteria is sensed in a TLR8- and RNase 6-dependent manner. RNase 6 showed cleavage at 3′ of uridines followed by adenosine or cytosine. The addition of a 2′O-Me at the uridine in the dinucleotide, UA—an RNase 6 cleavage site—prevented cleavage and, therefore, the generation of uridine-terminated fragments. 2′O-Me-modified ORNs showed a lack of TLR8 stimulation thus highlighting the critical role of RNase 6 for TLR8 activation and bacteria recognition.

## Results and discussion

### *RNASE6* is downregulated upon infection with bacteria recognized by TLR8

BLaER1 cells are a lymphoma B cell line transduced with a C/EBPα-ER-IRES-GFP construct enabling them to transdifferentiate into monocyte-like cells (Rapino et al, 2013) (Fig. 1A). Since these cells can be gene-edited and show TLR8 upregulation at the monocyte-like stage, we selected this cell line to investigate the innate immune response in *TLR8* knockout cells. Transdifferentiated BLaER1 cells were infected with various live whole bacteria at different multiplicities of infection (MOIs). We observed that IL-6

secretion was abrogated or severely diminished upon infection with *S. aureus* and *E. faecalis* in cells lacking TLR8 (Fig. 1B). We confirmed TLR8 dependency in primary human monocytes using CU-CPT9a, a TLR8 antagonist (Fig. EV1A).

Lack of TLR8 or blockage by CU-CPT9a was reported preventing the secretion of proinflammatory cytokines, such as IL-1β, TNF-α, IL-6, and CXCL10 upon infection with *Streptococcus agalactiae, S. aureus, S. pyogenes*, and *Pseudomonas aeruginosa* in primary cells as well as monocytic cell lines (Moen et al, 2019; Greulich et al, 2019; Ostendorf et al, 2020). TLR8-dependent induction of IFN-β release by monocytes has also been reported in response to *Borrelia burgdorferi* whole bacteria (Cervantes et al, 2011). A crosstalk between the TLR8 signaling pathway and cell surface TLR2, which is expressed at low levels in transdifferentiated BLaER1 cells, has been suggested (Moen et al, 2019), specifically for the recognition of *S. aureus*. Thus, culminating data show that TLR8 plays a physiological role in recognizing bacteria, despite the potential presence of different pathogen-associated molecular patterns (PAMPs). Therefore, our findings corroborate with previous work.

We next examined gene expression regulation of TLR8 in cells infected with living *S. aureus* and *E. faecalis* or treated with TL8-506, a potent TLR8 agonist. Additionally, we also analyzed the gene regulation of RNases which have been previously annotated as "lysosomal" such as RNase 6, RNase T2, and RNase 2 (Greulich et al, 2019; Ostendorf et al, 2020). All RNase genes were downregulated upon treatment with TL8-506 hinting to a possible mechanism of counterregulation. Yet, we cannot exclude other mechanisms. Live bacterial infection with both bacteria-induced downregulation of *RNASE6* while *E. faecalis* also induced downregulation of *TLR8* (Fig. 1C). Although the role of RNases in T cells is still unclear, downregulation of RNase 6 in response to HIV infection has been observed and suggests a protective role for RNase 6 in the immune response against pathogens (Christensen-Quick et al, 2016).

Since microbial RNA sensing by TLR8 takes place in the endolysosome, we performed a subcellular fractionation of the post-nuclear BLaER1 supernatant and analyzed it by mass spectrometry. This analysis revealed that RNase 6 and RNase T2 were enriched in the same cell fraction as TLR8 and TLR7 together with endolysosomal proteins, whereas RNase 2 expression was below the threshold of detection (Fig. EV1B). Sleat and colleagues (2013) have analyzed the subcellular distribution of RNase 6 within murine cells, reporting its colocalization with lysosomal markers (Sleat et al, 2013). RNase T2 has also been detected in the lysosome (Campomenosi et al, 2006). Therefore, we hypothesized that RNase 6 is potentially involved in the upstream processing of RNA within the endolysosome that also contains TLR8 (Itoh et al, 2011; Ishii et al, 2014).

### RNase 6 is required to generate TLR8 active RNA fragments from bacterial RNA

We next assessed the relative *RNASE6*, *RNASET2*, and *RNASE2* mRNA and protein expression in the human monocytic THP-1 cell line as well as BLaER1 cells at B-cell undifferentiated and monocyte-like transdifferentiated stages. The mRNA expression levels were compared to peripheral blood mononuclear cells (PBMCs) and CD14+ isolated monocytes. All RNases showed mRNA expression in primary cells, but none or low expression in

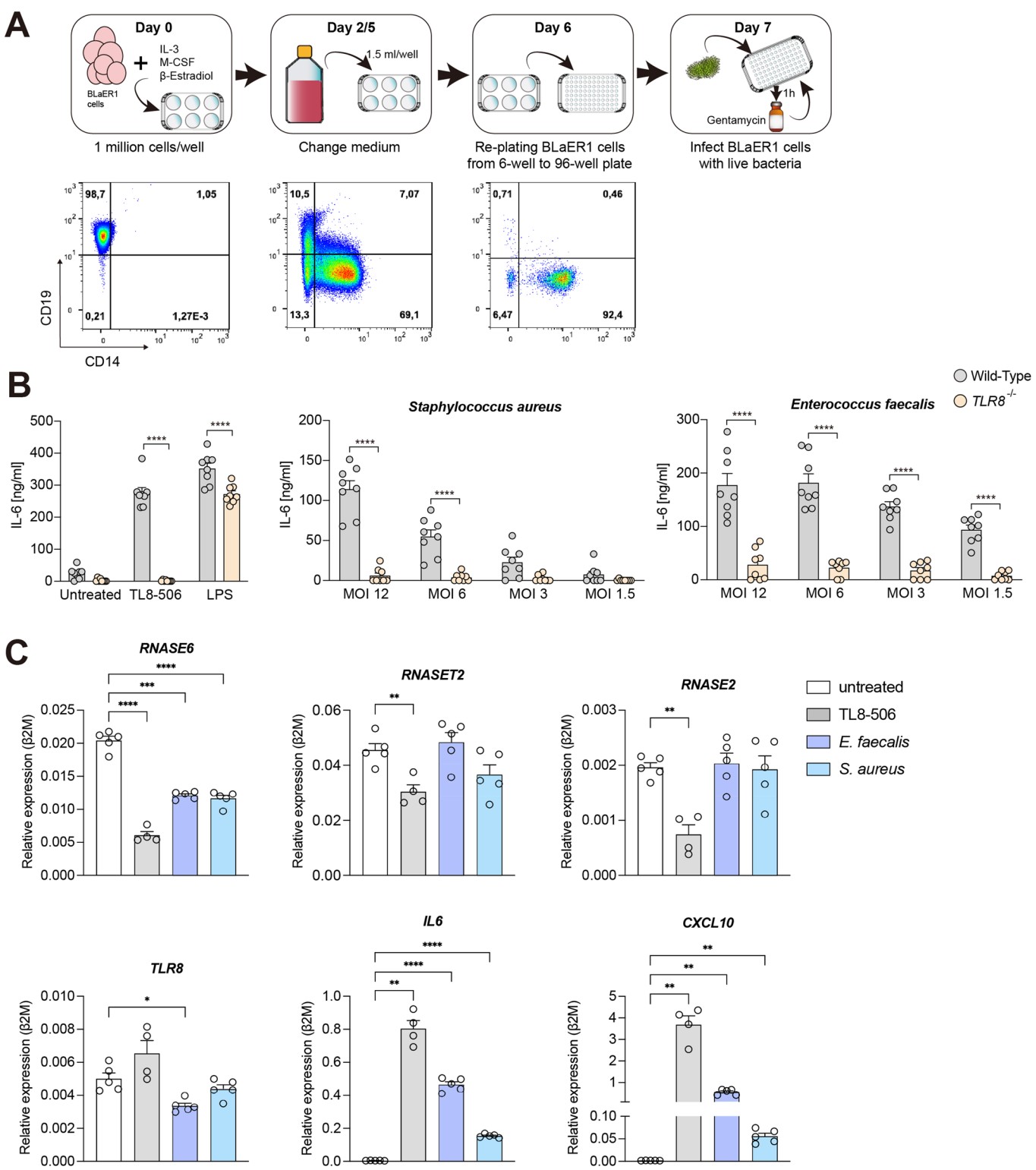

undifferentiated BLaER1 cells (Fig. 2A). *RNASE6* and *RNASET2* expression was upregulated in transdifferentiated BLaER1 cells which showed low *RNASE2* expression. On the other hand, THP-1 cells highly expressed *RNASE2* and *RNASET2*, whereas *RNASE6* could not be detected to that extent (Fig. 2A).

The detection of RNases at the protein level by western blot revealed similar findings for the cell lines (Fig. EV2A). CD14⁺ monocytes highly expressed RNase 6, followed by RNase 2 to a lesser extent, while RNase T2 showed the lowest expression (Fig. EV2A). Protein detection by western blot showed RNase 6

**Figure 1.** *RNASE6 is downregulated upon infection with TLR8-dependent bacteria.*

(A) Overview of live whole bacterial infection experimental design. BLaER1 cells were stained for CD19 and CD14, and analyzed by flow cytometry at day 0, 2, and 6 post induction of transdifferentiation. On day 7, cells were infected with live bacteria. To stop the infection, gentamycin (50 µg/ml) was added to the wells; then, cells were incubated for an additional 2 or 20 h. (B) BLaER1 cells Wild-Type or deficient for TLR8 were transdifferentiated and infected at different MOIs. Cell-free supernatant was analyzed for the secretion of IL-6 after 20-h incubation ($n = 4$, each with technical duplicates). Data were shown as mean ± SEM and was analyzed by a two-way ANOVA test followed by Šídák's multiple comparisons test. ****$p < 0.0001$. (C) BLaER1 cells were infected with MOI 20 of the respective strain or stimulated with TL8-506. Gentamycin was added to the wells, and after 2 h RNA was isolated to analyze mRNA expression. Data were normalized to β-2-microglobulin expression (β2M) (4–5 replicates). Data were shown as mean ± SEM and was analyzed by Brown–Forsythe and Welch ANOVA test followed by Dunnett's post-test. Exact $p$ values for individual sub-panels are reported here: *RNASE6*, ***$p = 0.0002$, ****$p < 0.0001$; *RNASET2*, **$p = 0.096$; *RNASE2*, **$p = 0.087$; *TLR8*, *$p = 0.0137$; *IL6*, **$p = 0.0012$, ****$p < 0.0001$; *CXCL10*, **$p = 0.0067$, 0.0016, or 0.0033, comparing TL8-506, *E. faecalis*, or *S. aureus* to untreated group, respectively. Source data are available online for this figure.

detection in BLaER1 cells and THP-1. Moreover, mass spectrometry confirmed RNase 6 detection in BLaER1 cells (Fig. EV2B). Overall, these findings follow those observed in previous studies (Ostendorf et al, 2020; Tong et al, 2023), yet arguing for RNase 6 being more important for RNA processing upstream of TLR8 activation since CD14$^+$ monocytes are characterized as highly RNase 6 expressing cells and this was best mirrored in transdifferentiated BLaER1 cells. Based on these findings, we genomically edited BLaER1 cells by CRISPR-Cas9 to lack RNase 6 which was confirmed by antibody detection and mass spectrometry (Figs. 2B and EV2C).

As *S. aureus*, and *E. faecalis* showed strong TLR8-dependent recognition (Fig. 1B) we hypothesized that the lack of RNase 6 might impair the recognition of bacteria in a similar way as previously reported for RNase T2 with *S. aureus* (Greulich et al, 2019). We infected cells with these two bacteria to assess the physiological role of RNase 6 and measured proinflammatory cytokines in the cell-free supernatant from BLaER1 cells lacking *RNASE6*. Live whole bacterial infection with *S. aureus* in *RNASE6*$^{-/-}$ cells showed a strong and significant reduction of IL-6 and TNF-α secretion (Figs. 2C and EV2D). Despite the strong dependency on TLR8 recognition (Fig. 1B), live infection with *E. faecalis* in RNase 6 deficient cells only resulted in a significant, yet partial reduction of cytokine secretion at the highest MOI. In a recent experimental model of urinary tract infection, Cortado and colleagues (Cortado et al, 2024) observed that bone marrow-derived monocytes from *Rnase6* deficient mice presented reduced uropathogenic *E. coli* intracellular killing. Together, these results suggest an important role for RNase 6 against bacterial infection.

Analysis of the breakdown fragments by agarose gel electrophoresis after bRNA incubation with recombinant RNase 6 showed that the digestion of bRNA from both strains to shorter fragments occurred in a concentration-dependent manner (Fig. EV2E). Next, BLaER1 cells were transfected with total isolated bRNA from both strains, and cytokines were measured. The experiment revealed an RNase 6-dependent IL-6 response upon stimulation with bRNA (Figs. 2D and EV2F). Of note, this phenotype in BLaER1 cells was again markedly stronger for *S. aureus* total RNA, which has previously been shown to rely on TLR8 sensing in primary monocytes and monocyte-derived macrophages and showed dependency on RNase T2 or both, RNase T2 and RNase 2 processing (Bergstrøm et al, 2015; Greulich et al, 2019; Ostendorf et al, 2020). Although the dependence of *E. faecalis* RNA on TLR8 sensing has previously been shown (Nishibayashi et al, 2015), RNase dependence on this matter has not been previously assessed. Lack of RNase 6 affected the IL-6 response upon stimulation at a lower RNA concentration, but this effect was not sustained at a

higher concentration of bRNA. We hypothesize this phenotype might be due to the effect of the remaining activity of other expressed RNases in the system, bRNA sequence composition or even RNA modifications which might limit RNase activity.

Next, we evaluated the immunostimulatory potential of fragments generated by RNase 6 cleavage. We digested *E. faecalis* RNA with recombinant RNase 6, and thereafter, the reaction products were transfected into BLaER1 cells. IL-6 measurements showed a 1.4-fold increase in wild-type cells, arguing that RNase 6 breakdown fragments are potentially immunostimulatory, sensed by TLR8, and can even increase stimulation in RNase 6 competent cells. More importantly, a 2.5-fold increase in IL-6 production was observed in cells lacking RNase 6, ultimately rescuing in part their phenotype and proving that *ex cellulo* RNase 6 digested RNA fragments are active in a TLR8-dependent manner (Figs. 2E and EV2G). The stimulatory potential of breakdown products has been indicated by previous studies using a similar digestion approach applied to synthetic oligoribonucleotides known to induce a potent TLR8 response. For instance, pretreatment of RNA40 (5′-GCCCGUCUGUUGUGUGACUC-3′) with RNase T2 could rescue the phenotype in *RNASET2*$^{-/-}$ cells (Greulich et al, 2019). Similarly, predigestion of ORN002 (5′-(UUA)$_6$UU-3′) with RNase T2 and RNase 2 could also marginalize the phenotype in *RNASET2*$^{-/-}$ *RNASE2*$^{-/-}$ cells (Ostendorf et al, 2020). Taken together, these findings emphasize the relevance of RNA processing for the generation of motifs that can engage TLR8.

To test if these findings are also valid in primary human cells, we next assessed the RNase6-mediated response in CD14$^+$ monocytes that showed high RNase 6 expression. Monocytes were edited by CRISPR-Cas9 using several single guide RNA (sgRNA) sequences targeting *RNASE6* or a non-targeting scrambled sgRNA. After cultivation for 6 days, Western blot analysis revealed that RNase 6 was undetectable in sg*RNASE6* edited cells (Fig. 2F). Cells were then stimulated with RNA from *E. faecalis* or *S. aureus*. TNF-α production was significantly decreased in monocytes lacking RNase 6 upon stimulation (Fig. 2G), yet in monocytes *S. aureus* RNA showed reduced dependency. To confirm findings for *S. aureus*, additional concentrations were tested, all showing a significant reduction of cytokine secretion in monocytes lacking RNase 6 (Fig. EV2H). Of note, predigestion with recombinant RNase 6 before the transfection induced a rescue of TNF-α stimulation to the basal level of non-targeted stimulated cells (Fig. 2H,I). In addition, predigestion of bRNA also induced higher TNF-α production in non-targeted cells (Fig. 2I). The results confirm the dependency on RNase 6 for the processing and TLR8 stimulation by bRNA in primary monocytes.

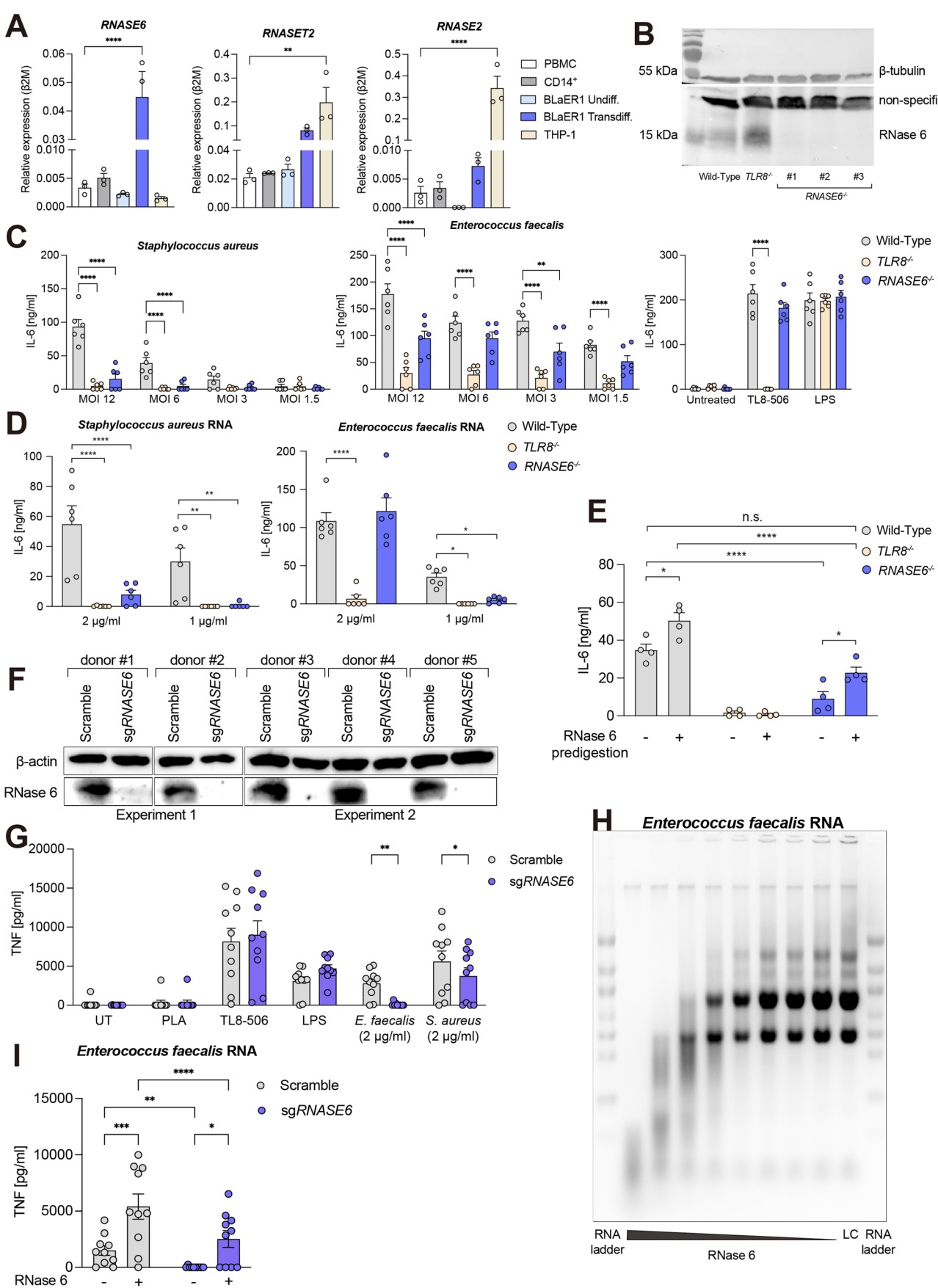

**Figure 2. RNase 6 is required to generate TLR8 active RNA fragments from bacterial RNA.**

(A) *RNASE6*, *RNASET2*, and *RNASE2* mRNA expression were analyzed in untreated PMBC, isolated CD14[+] monocytes, undifferentiated and transdifferentiated BLaER1, as well as THP-1 cells. Data were normalized to β-2-microglobulin (β2M) ($n = 3$ donors or three independent cell line passages). Data were shown as mean ± SEM and was analyzed by one-way ANOVA followed by Dunnett's post-test. Exact $p$ values are **$p = 0.0047$ and ****$p < 0.0001$. (B) Immunoblot for RNase 6 expression of wild-type or BLaER1 cells edited by CRISPR-Cas9. (C) BLaER1 cells were infected with whole live bacteria at different MOI. IL-6 was detected by ELISA upon live whole bacterial infection with *S. aureus* or *E. faecalis* at several MOIs. Controls include TL8-506 and LPS for stimulation ($n = 3$, each with technical duplicates). Data were shown as mean ± SEM and was analyzed by two-way ANOVA followed by Dunnett's test. Exact $p$ values are **$p = 0.0016$ and ****$p < 0.0001$. (D) BLaER1 cells were stimulated with isolated bRNA for 20 h and IL-6 was measured in the supernatant by ELISA ($n = 3$, each with technical duplicates). Data were shown as mean ± SEM and was analyzed with two-way ANOVA followed by Dunnett's test. Exact $p$ values for individual sub-panels are reported here: *Staphylococcus aureus* RNA, ****$p < 0.0001$, **$p = 0.0046$ and 0.0056; *Enterococcus faecalis* RNA, ****$p < 0.0001$, *$p = 0.0156$ and 0.0407, comparing *TLR8*[−/−] or *RNASE6*[−/−] to wild-type group at indicated concentration, respectively. (E) 10 μg of isolated *E. faecalis* was digested with 1 ng of recombinant RNase 6 (+). Breakdown fragments were transfected (1 μg/ml) into BLaER1 cells and IL-6 was measured after 20 h of incubation ($n = 2$, each with technical duplicates). Data is shown as mean ± SEM and was analyzed by two-way ANOVA followed by Tukey's multicomparison test. Exact $p$ values for comparing the effect of RNase 6 predigestion in Wild-Type or *RNASE6*[−/−] are *$p = 0.0155$ or *$p = 0.0369$, respectively; comparing genetically loss of RNASE6 are ****$p < 0.0001$ and not significant (n.s.) $p = 0.0931$. (F) Immunoblots for RNase 6 detection in lysates of primary human CD14[+] monocytes from five different donors (5 μg of protein) after 6 days of cultivation post genomic edition by CRISPR-Cas9. (G) CD14[+] monocytes edited by CRISPR-Cas9 shown in (F) were transfected with 2 μg/ml of undigested bRNA. TNF-α was measured after 20-hour incubation. ($n = 2$ in a total of 5 donors, each with technical duplicates). Data were shown as mean ± SEM and was analyzed by two-way ANOVA followed by Šídák's multiple comparisons test. The exact $p$ values are **$p = 0.0045$ or *$p = 0.0174$. (H) 10 μg of *E. faecalis* RNA was incubated with different amounts of recombinant RNase 6 (16 to 0.1 ng). Degradation products were run in 1X TAE 1.5% agarose gel to visualize the RNA fragmentation. Representative image out of two gels. (I) Fragments out of *E. faecalis* RNA digestion by RNase 6 (1 ng) were transfected into the cells at 1 μg/ml, and TNF-α was measured after 20 h of incubation. Undigested *E. faecalis* RNA was also transfected into the cells at the same concentration as a control ($n = 2$ in a total of five donors, each with technical duplicates). Data were shown as mean ± SEM and was analyzed by two-way ANOVA followed by uncorrected Fisher's LSD test. Exact $p$ values are *$p = 0.0176$, **$p = 0.0086$, ***$p = 0.0004$, and ****$p < 0.0001$. pLArg poly-L-arginine, LPS lipopolysaccharide, sg*RNASE6* single guide *RNASE6*. Source data are available online for this figure.

## RNase 6 cleavage releases uridine-terminated RNA fragments

Transferase-type ribonucleases, including RNase 6 and RNase T2, cleave the phosphodiester bond of RNA (Russo et al, 2001; Luhtala and Parker, 2010). At their enzymatic catalytic core, the phosphate-binding sites (P0, P1, and P2) anchor the RNA backbone while the nucleotides are recognized at the base-binding sites B1 and B2 according to their selectivity (Fig. 3A) (Russo et al, 2001). It has been reported that RNase 6 selectively recognizes pyrimidines at the B1 site (Pulido et al, 2016; Prats-Ejarque et al, 2016) whereas RNase T2 selectively cleaves in between purines (B1 site) and uridine (B2 site) (Greulich et al, 2019). Initially, a transphosphorylation at the phosphodiester bond between the P1 site and the B2 site generates a 2′3′-cyclophosphate-terminated intermediate, which may eventually undergo hydrolysis and turn into a 3′-phosphate-terminated fragment.

As RNase 6 breakdown fragments showed strong TLR8 stimulation, we tested RNase 6 cleavage on a UUNN ribonucleotide motif embedded in non-stimulatory deoxynucleotide sequences, (dAdC)$_3$UUNN(dAdC)$_4$, in an ex cellulo digestion assay (Fig. 3B). We visualized the generated fragments in urea-denaturing gels and observed that RNase 6 efficiently cleaved UUAA, but it was ineffective against the UUGG sequence (Fig. 3C). A similar cleavage pattern was observed after UpA (UA dinucleotide with phosphate backbone) and UpG dinucleotides cleavage (Prats-Ejarque et al, 2016). UUUU and UUCC were also susceptible to RNase 6 cleavage (Fig. 3C). Our findings corroborate with the observations from Tong and colleagues (Tong et al, 2023) that have recently shown cleavage between UC by RNase 6 in pUC.

Of note, no cleavage was observed for any of the UUNN chimeric sequences after incubation with RNase T2 (Fig. 3D). To further assess the cleavage specificity of RNase 6 and RNase T2, we designed GGNN also flanked by non-stimulatory deoxynucleotide sequences (Fig. EV3A). As predicted, RNase T2 was able to cleave only GGUU (Fig. EV3B) which is consistent with previous data showing preferential RNase T2 cleavage between GU-releasing

guanosine-terminated fragments for TLR8 concave site (Greulich et al, 2019).

Upon cleavage by RNase 6, GGUU and GGCC sequences were degraded (Fig. EV3C). RNase A family members and RNase T2 show contrasting base preferences for their site of cleavage. Still, we observed degradation of GGUU by both RNase T2 and RNase 6. Therefore, we focused on deciphering where RNase 6 could cleave on GGUU. We noticed that there was a rUdA site on the (dAdC)$_3$GGUU(dAdC)$_4$ sequence which could perhaps serve as a site cleavage for RNase 6. Therefore, we engineered the (dAdC)$_3$U(dAdC)$_4$ sequence (Fig. 3A) which was degraded by RNase 6 as well, but not by RNase T2 (Fig. 3E). Similarly, a fluorescent substrate 6-FAM~dArUdAdA~6-TAMR showed cleavage by RNase 7, another RNase A family member (Rademacher et al, 2019). These findings suggest that RNase 6 is able to cleave when uridine is positioned at the B1 site and followed by deoxy adenosine in chimeric sequences, also arguing that the cleavage on (dAdC)$_3$UUUU(dAdC)$_4$ and (dAdC)$_3$GGUU(dAdC)$_4$ might derivate from the provided rUdA. Based on previous data, it is unlikely that RNase 6 and RNase T2 are preferentially cleaving at the same position in those sequences (GU or GC). A similar approach could be applied to pin down the site cleavage in GGCC. Of note, RNase 6 has shown catalytic activity on CpA (Prats-Ejarque et al, 2016). However, C-terminated sequences have not shown a potential effect on TLR8 stimulation, and, therefore, this sequence was not further analysed in this study. Taken together, these findings highlight that RNase 6 generates uridine-terminated fragments and, in collaboration with RNase T2 – which releases guanosine-terminated products—can activate TLR8.

## RNA 2′O-methylation between uridine and adenosine impairs RNase 6 cleavage and TLR8 response

Although the base preferences may indicate a preferential site for cleavage, the transphosphorylation takes place on the RNA backbone, suggesting that structural RNA modifications may directly affect RNase cleavage. Indeed, a broad range of

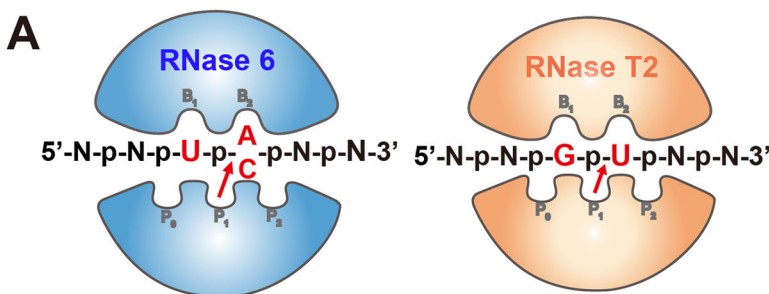

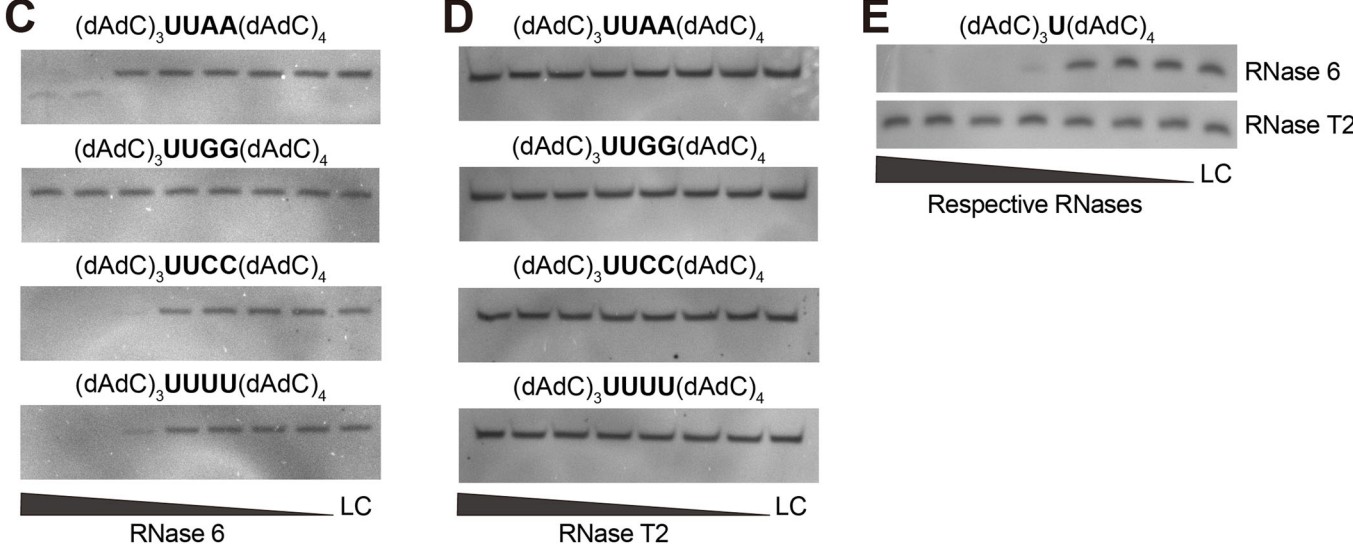

**Figure 3. RNase 6 cleavage releases uridine-terminated RNA fragments.**

(A) Scheme of enzymatic center of transferase-type RNases. The red arrow indicates the site of transphosphorylation. The scheme is based on Russo and colleagues (2011). (B) Table of chimeric ORNs (protected deoxynucleotide sequences with central RNA) used for *ex cellulo* digestion by RNase 6 and RNase T2. (C) Urea gel of (dAdC)$_3$UUNN(dAdC)$_4$ digested by recombinant human RNase 6 or (D) RNase T2 (5 to 0.05 ng/µl). An untreated sample was loaded as a Loading Control (LC). (E) Urea gel of (dAdC)$_3$U(dAdC)$_4$ digested by recombinant human RNase 6 or RNase T2 (5 to 0.05 ng/µl). An untreated sample (LC) was loaded as a control. (C–E) Representative image out of two gels. N nucleotide, d deoxynucleotide, r ribonucleotide, A adenosine, G guanosine, U uridine, C cytosine). Source data are available online for this figure.

posttranscriptional base and 2′ sugar modifications have been identified in RNA (reviewed in (Freund et al, 2019b) with respect to immunological effects). Structural RNA modifications are naturally abundant in host RNA, playing a crucial role in avoiding self-recognition by TLRs (Karikó et al, 2005). Despite being less frequent in microbial RNA, modifications may have evolved there as a strategy to escape from host recognition (Galvanin et al, 2020). Amongst the several posttranscriptional ribose modifications, 2′O-Me shows immunosuppressive properties on TLRs, even when added in trans to a stimulatory RNA (Schmitt et al, 2017). Ostendorf and colleagues (Ostendorf et al, 2020) showed the

complete impairment of RNase 2 and T2 cleavage by the insertion of 2′O-Me in the full length of a synthetic ORN. Recently, 2′O-Me- and 2′fluoro-U-modified synthetic ORNs were shown to prevent RNase 6 cleavage in UC-rich sequences (Tong et al, 2023).

Since UA has been proposed as a cleavage site for RNase 6, we asked whether a single 2′O-Me-U could fully impair RNase 6 activity. To address this question, we designed new UUAA chimeric sequences where a 2′O-Me was added to each individual nucleotide, resulting in the following additional variants: UmUAA, UUmAA, and UUAmA (Fig. 4A). Next, these synthetic sequences were *ex cellulo* digested by RNase 6 as previously described. Analysis of the

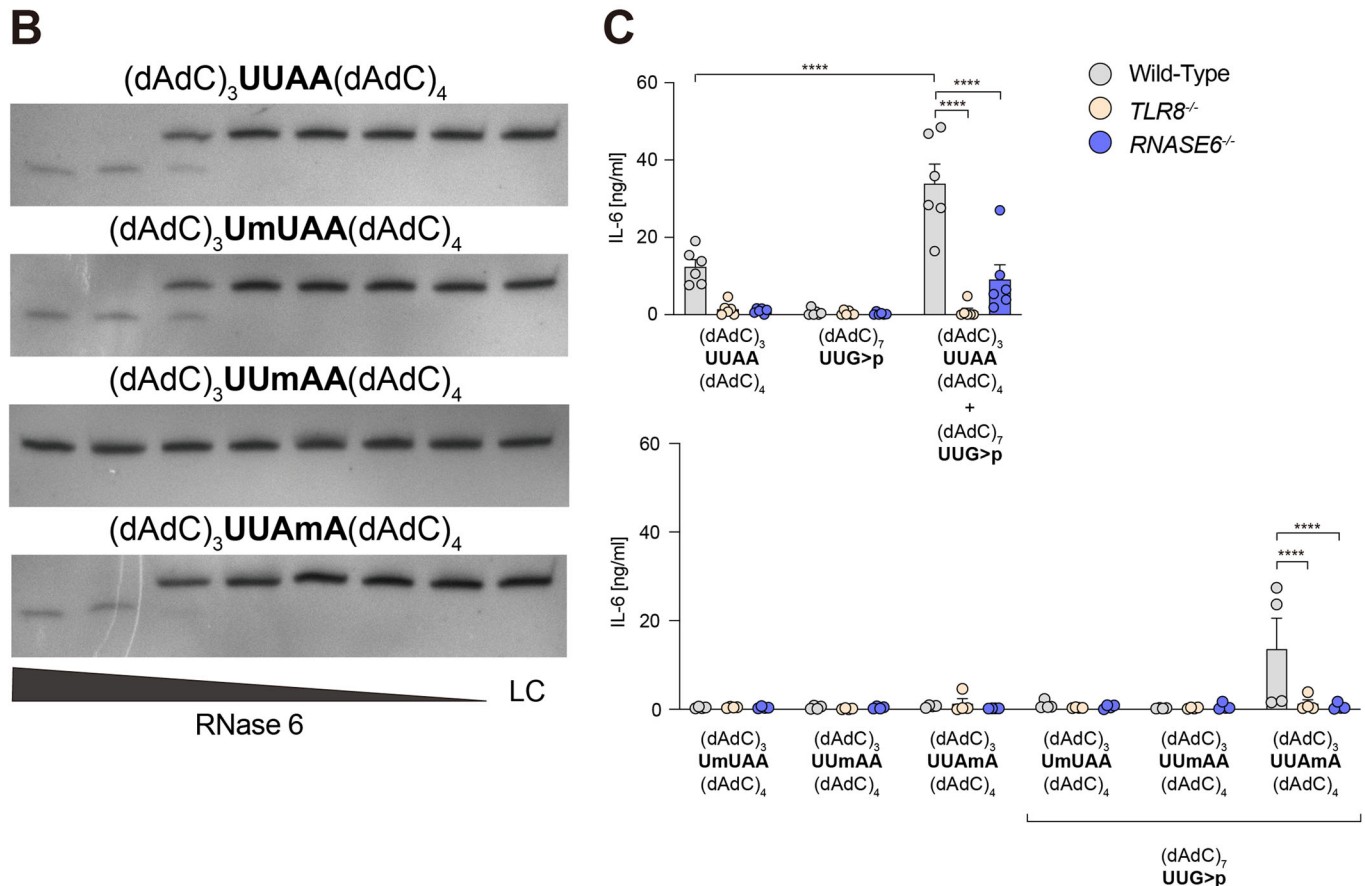

**Figure 4. RNA 2′-O-methylation between Uridine and Adenosine impairs RNase 6 cleavage and TLR8 response.**

(A) Table of un- and modified chimeric ORNs used for *ex cellulo* digestion by recombinant human RNase 6 (d deoxynucleotide, r ribonucleotide, A adenosine, G guanosine, U uridine, C cytosine, m 2′-O-methylation; >p, 2′3′-cyclophosphate termination). (B) Urea gel of (dAdC)$_3$UUAA(dAdC)$_4$ and modified variants digested by recombinant human RNase 6 (5 to 0.05 ng/µl). An untreated sample was loaded as a control (LC, loading control). Representative image out of two gels. (C) IL-6 measurements after stimulation with chimeric ORNs alone (4.8 µg/ml) or a combination of (dAdC)$_3$UUAA(dAdC)$_4$ or its modified variants and (dAdC)$_7$UUG > p (4.8 µg/ml each), an RNase T2-predicted fragments ($n = 2$–3, each with technical duplicates). Data were shown as mean ± SEM and was analyzed with two-way ANOVA followed by Dunnett's test. Exact $p$ values are ****$p \le 0.0001$. Source data are available online for this figure.

breakdown products revealed that 2′O-Me in UUmAA blocked RNase 6 cleavage whilst the modification in all the other modified variants did not affect the generation of similar fragments as the unmodified oligonucleotide (Fig. 4B). Thus, this observation suggests UA as a potential preferential RNase 6 cleavage site.

The chimeric UUAA oligonucleotide has been described as a uridine donor giving rise to the ligand for the apex site in TLR8, as high IL-6 levels were detected upon simultaneous transfection of UUAA

with (dAdC)$_7$UUG > p, from now on referred to as UUG>p, a predicted RNase T2 breakdown product (Greulich et al, 2019). Considering the crucial role of uridine residues in the TLR8 apex binding site, we aimed to analyze the impact of the modified UUAA variants on TLR8 activation. Therefore, we co-stimulated transdifferentiated BLaER1 cells with all designed UUAA-modified variants and predicted RNase T2 fragment (Fig. 4A). High IL-6 levels were detected in response to UUAA + UUG>p combination, and this response was

RNase 6-dependent. Although IL-6 release was induced to a lesser extent, the modified variant UUAmA showed a similar phenotype (Fig. 4C). Notably, UA-rich sequences were previously shown to selectively activate TLR8 (Forsbach et al, 2008).

Of note, the IL-6 response vanished upon co-stimulation with UUmAA and UUG>p, indicating that even in the presence of RNase T2-predicted fragments, TLR8 stimulation was still dependent on RNase 6 processing. Accordingly, the addition of 2′O-Me at the uridine in pUC also drastically affected the generation of fragments as the modification prevented RNase 6 cleavage (Tong et al, 2023). Our group has demonstrated that 2′O-Me-G within bacterial transfer-RNA (tRNA) exerts a potent antagonism towards TLR7/8 activation inhibiting TNF-α secretion by otherwise stimulatory total bacterial RNA (Rimbach et al, 2015). Likewise, the knockout of tRNA methyltransferase H (trmH) in *E. coli* elicited significantly higher levels of IFN when compared to the wild-type bRNA (Freund et al, 2019a). Taken together, these findings highlight the relevance of RNase 6 on the upstream RNA processing and also reveal that 2′O-Me-U downstream followed by A in bRNA might be relevant for TLR8 sensing.

Strikingly, UmUAA also abrogated TLR8 response (Fig. 4C), although this modified variant was still cleaved by RNase 6, and the breakdown fragments are similar to those of the unmodified sequence (Fig. 4B). Although the mechanism of 2′O-Me antagonistic effect is not fully elucidated for TLR8, stronger binding of 2′O-Me-modified RNA to TLR7 has been already shown (Hamm et al, 2010). Since 2′O-Me-modified fragments (UmU) may still be produced by RNase 6, we hypothesize that site-specific 2′O-Me in bRNA may dampen TLR8 response when: (1) placed at the RNase preferential site and impairing cleavage and/or (2) located in the upstream neighboring positions to RNase cleavage resulting in modified-containing U-terminated fragments.

In summary, the data presented in this study proves the importance of RNase 6 for TLR8 stimulation by bRNA, thus adding a physiological function to RNase 6 for innate activation within monocytes. RNase 6 shows an essential contribution to bacteria recognition, cleaving after uridines and generating uridine residues for TLR8 sensing. 2′O-Me placed at the uridine in the dinucleotide sequence UA showed impairment of cleavage by RNase 6 and subsequent loss of TLR8 stimulation. Additional mechanistic studies will be required to understand 2′O-Me-modified RNA binding activity on TLR8. Our findings contribute to a better understanding of bRNA/TLR8 activation and help to develop approaches to block TLR8 activation.

# Methods

### Reagents and tools table

| Reagent/resource | Reference or source | Identifier or catalog number |
|---|---|---|
| **Experimental models** | | |
| BLaER1 B-cell precursor leukemia cell line (*Homo sapiens*) | Prof. Dr. Holger Heine (Research Center Borstel, Germany) | |
| THP-1 cell line (*Homo sapiens*) | Prof. Dr. Axel Roers (University Heidelberg, Germany) | |

| Reagent/resource | Reference or source | Identifier or catalog number |
|---|---|---|
| Human peripheral blood mononuclear cells | Buffy coats from IKTZ Heidelberg | |
| Isolated primary CD14+ human monocytes | Buffy coats from IKTZ Heidelberg | |
| *Staphylococcus aureus* | DSMZ | ATCC 25923 |
| *Enterococcus faecalis* | DSMZ | ATCC 29212 |
| **Recombinant DNA** | | |
| pU6-(BsbsI)_CBh-Cas9-T2A-BFP plasmid | Addgene | 64323 |
| **Antibodies** | | |
| Monoclonal Rabbit anti-β-Tubulin | Cell Signaling Technology | 2128 |
| Mouse anti-β-Actin | Cell Signaling Technology | 3700S |
| Polyclonal Rabbit anti-Human RNASE6 | LifeSpan Biosciences | LS-C726376-100 |
| Polyclonal Rabbit anti-RNASET2 | Sigma-Aldrich | HPA029013-100UL |
| Polyclonal Rabbit anti-RNASE2 | Thermo Fisher | PA5-97305 |
| Goat anti-Rabbit IgG secondary antibody | Cell Signaling Technology | 7074S |
| Horse anti-Mouse IgG secondary antibody | Cell Signaling Technology | 7076S |
| PE anti-human CD19 | BioLegend | 392506 |
| APC anti-human CD14 | BioLegend | 367118 |
| **Oligonucleotides and other sequence-based reagents** | | |
| sgRNASE6 | This study | Appendix Table S1 |
| PCR primers | This study | Appendix Table S2 |
| Chimeric oligoribonucleotides | biomers.net | Main figures |
| **Chemicals, enzymes and other reagents** | | |
| PageRulerTM Plus Prestained Protein Ladder | Thermo Fisher | 26619 |
| Westar ηC Ultra 2.0 | Cyanagen | XLS075 |
| Pancoll human | Pan-Biotech | P04-60500 |
| β-estradiol | Sigma-Aldrich | E2758 |
| RPMI 1640 with stable glutamine | Anprotech | AC-LM-0056 |
| Recombinant Human IL-3 | Peprotech | AF-200-03 |
| Recombinant Human M-CSF | Peprotech | AF-300-25 |
| NEB® 5-alpha *E. coli* | New England Biolabs | C2987 |
| EndoFree Plasmid Maxi kit | Qiagen | 12362 |
| Lipofectamine™ 3000 / P3000 reagent | Invitrogen | L3000001 |
| Opti-MEM® Reduced serum medium | Gibco | 31985-062 |
| P3 Primary Cell 4D-Nucleofector X kit L | Lonza | LONV4XP-3024 |

| Reagent/resource | Reference or source | Identifier or catalog number |
|---|---|---|
| *SpCas9 Nuclease* | IDT | 1081059 |
| RNeasy Plus Kit | Qiagen | 74134 |
| RevertAid First Strand cDNA Synthesis Kit | Thermo Fisher | K1622 |
| SYBR™ Green PCR Master Mix | Applied Biosystems | 4309155 |
| SDS (Sodium lauryl sulfate) | Carl Roth | 4360.1 |
| NuPAGE LDS sample buffer | Thermo Fisher | NP0007 |
| Coomassie brilliant blue G250 | Carl Roth | 9598.1 |
| DL-Dithiothreitol | Merck | D0632-25G |
| Iodoacetamide | Merck | I6125 |
| MS-grade trypsin | SERVA | 37286 |
| StageTips | Cole Parmer | GZ-35213-24 |
| Aurora ultimate column | Ionopticks | AUR3-25075C18 |
| HEPES/KOH pH 7.6 | Carl Roth | HN78.2 |
| Magnesium chloride | Sigma-Aldrich | M2670 |
| Potassium chloride | Carl Roth | 6781.1 |
| Igepal CA-630 | Sigma-Aldrich | I8896 |
| cOmplete Protease Inhibitor Cocktail | Roche | 54925800 |
| Sodium chloride | Sigma-Aldrich | 71380 |
| EDTA | Carl Roth | 8040.2 |
| Sucrose | Sigma-Aldrich | S7908-250g |
| Pierce™ BCA Assay Kit | Thermo Fisher | 23225 |
| 4–20% Tris-Glycin-Gel 1,0 mm | Anamed | TG 42012 |
| PageRuler™ Prestained Protein Ladder | Thermo Fisher | 26619 |
| Blue Block | SERVA | 42591 |
| Lysozyme from chicken egg white | SERVA | 28263.02 |
| TRIzol reagent | Thermo Fisher | 15596018 |
| CU-CPT9a | Invivogen | inh-cc9a |
| Recombinant human RNase 6 | Creative Biomart | RNASE6-184H |
| Recombinant human RNase T2 | Cusabio | CSB-MPO19810HU |
| Tris-EDTA buffer pH 8.0 | Sigma-Aldrich | 93283 |
| 2X RNA Loading Dye | Thermo Fisher | R0641 |
| Diethyl pyrocarbonate (DEPC) | VWR | A0881 |
| pegGREEN, DNA/RNA dye | peqlab | 732-3196 |
| 10X TBE | SERVA | 3932.01 |
| poly-L-arginine | Sigma-Aldrich | P7762 |
| TL8-506 | InvivoGen | tlrl-tl8506 |

| Reagent/resource | Reference or source | Identifier or catalog number |
|---|---|---|
| LPS | InvivoGen | Tlrl-peklps |
| IL-6 BD OptEIA™ | BD Biosciences | 555220 |
| TNF BD OptEIA™ | BD Biosciences | 555212 |
| **Software** | | |
| GraphPad Prism 10 | https://www.graphpad.com/ | |
| MaxQuant (2.1.0.0) | https://www.maxquant.org/ | |
| **Other** | | |
| Biometra Fastblot ™ | Analytik Jena | 846-015-200 |
| autoMACS® | Miltenyi Biotec | |
| MACSQuant 10 | Miltenyi Biotec | |
| BD FACSAria™ | BD Biosciences | |
| 4D-Nucleofector | Lonza | |
| timsTOF Pro | Brucker | |
| Easy-nLC 1200 | Thermo Fisher | |
| Orbitrap Exploris 480 mass spectrometer | Thermo Fisher | |
| Dounce tissue grinder with 0.071 mm clearance | DKW Life Sciences | 8853000007 |
| Ultra-clear tube | Beckman Coulter | |
| Tecan microplate reader | TECAN | |
| PerfectBlue™ Vertical Double Gel System | peqlab | |
| Intas ChemoStar imager | Intas | |
| NanoDrop ND-1000 Spectrophotometer | Marshall Scientific | |
| nanoElute HPLC | Bruker | |

## Human resources

Human peripheral blood cells used in the present study followed the official regulations and were approved by the local ethical committee (EK91022019, S-716/2017).

## Isolation of PBMC and CD14$^+$ cells

Buffy coats from healthy donors were used for PBMC isolation by gradient centrifugation. Initially, samples were diluted one to threefold with 1X PBS. To form a density phase separation, 30 ml of diluted blood was carefully transferred to a 50 ml tube previously filled with 15 ml Pancoll (PAN-BIOTECH) and centrifuged at $900 \times g$ at room temperature for 20 min using switched-off brakes. The lymphocytes/PBMC ring was collected, mixed with 50 ml of 1X PBS, and centrifuged again at $400 \times g$ at room temperature for 10 min. This last washing step was repeated twice, and cells were finally collected. A portion of PBMC was subjected to CD14$^+$ isolation by magnetic cell sorting autoMACS® (Miltenyi Biotec).

## Cell lines

BLaER1 and THP-1 cells were cultivated in RPMI 1640 medium (Anprotec), supplemented with 10% FCS and 1% Penicillin/ Streptomycin (v/v), and subcultured every 4–5 days in T75 flasks. To induce transdifferentiation of BLaER1 cells into monocyte-like cells, 150 nM β-estradiol (Sigma-Aldrich), 10 ng/mL of the recombinant human growth factors IL-3, and M-CSF (Peprotech) were added to the medium and cells were cultured for 5–7 days in six-well plates. Transdifferentiation efficiency was accessed by PE anti-human CD19 and APC anti-human CD14 (BioLegend) cell surface staining and analysed by MACSQuant 10 (Milteny Biotec). For infection and stimulation experiments, transdifferentiated cells were seeded onto 96-well plates at a $2 \times 10^5$ cells/well density in 50 µl while $1 \times 10^6$ were seeded onto six-well plates in 3 ml for the 3 h of infection for gene expression analysis.

## BLaER1 genomic edition by CRISPR-Cas9

Procedures refer to the protocol published by Ran and colleagues (2013). Briefly, suitable single guide RNA (sgRNA) sequences for *RNASE6* were generated using the Benchling online software (https://benchling.com/). Scored sgRNAs are described in Appendix Table S1. To allow their cloning into the plasmid pU6-(BbsI) _CBh-Cas9-T2A-BFP (Addgene), the forward and reverse sgRNAs were flanked by a 5′ CACCG-sgRNA 3′ and a 3′ C-sgRNA-CAAA 5′ additional sequence, respectively.

Followed by transformation into NEB® 5-alpha *E. coli* (New England Biolabs) by heat shock, pU6-(BbsI)_CBh-Cas9-T2A-BFP-sgRNA was expanded, and isolated using Maxiprep kits (Qiagen). Before transfection, $1 \times 10^6$ undifferentiated cells were seeded onto six-well plates in an incomplete RPMI medium. For transfection, the plasmid was complexed with Lipofectamine™ 3000 (Invitrogen) following the manufacturer's instructions. Briefly, 4 µg of plasmid was mixed with P3000 reagent in Opti-MEM (gibco), and complexed with 3.75 µl of Lipofectamine previously prepared in Opti-MEM for 15 min at room temperature. Finally, 250 µl of DNA-Lipofectamine complexes were loaded on the well and incubated for 48 h at 37 ℃. Single GFP + BFP+ cells were sorted by BD FACSAria™ (BD Biosciences) onto 96-well plates, which were maintained at 37 ℃ for up to 3–4 weeks, followed by the genotyping of generated clones by Sanger Sequencing and Western Blot. TLR8 knockout cells were done in the same way and reported earlier (Vierbuchen et al, 2017).

## Nucleofection-based CRISPR-Cas9 gene-editing in primary human CD14+ monocytes

Following a recently established workflow for gene-editing in primary CD4 T cells (Morath et al, 2024; Albanese et al, 2022), we used a similar strategy for the knockout of *RNASE6* in CD14+ monocytes. Briefly, freshly isolated CD14+ monocytes ($10 \times 10^6$) were washed twice with PBS and resuspended in 100 µL P3 solution from the P3 Primary Cell 4D-Nucleofector X Kit L (Lonza). sgRNAs (designed and synthesized from Synthego) were incubated together with recombinant *SpCas9 Nuclease* (IDT) for 10 min at room temperature at a ratio of 1:2.5 (40 pmol Cas9 protein per 100 pmol sgRNA) to form the CRISPR-Cas9-sgRNA ribonucleoproteins (RNP) complex. For efficient KO of *RNASE6*, we used

three different sgRNAs targeting exon 2 of the gene in the same reaction (Appendix Table S1). About 10 µL of RNP complex per sgRNA and $10 \times 10^6$ cells were used per nucleofection reaction. Cells were nucleofected using the program EH-100 on the 4D-Nucleofector system. About 500 µL of pre-warmed RPMI (w/o supplements) was added, and cells were allowed to recover for 10 min at 37 °C. Subsequently, the culture medium was supplemented with the corresponding donor serum at 20%, and $5 \times 10^6$ cells were seeded onto 96-well plates ($2 \times 10^5$ cells/well) for stimulations, whereas the other $5 \times 10^6$ cells were split onto six-well plates for RNase 6 detection. Cells were stimulated with bRNA or lysed for protein detection 6 days post nucleofection.

## Cell lysate for mRNA and protein isolation

Regardless of the cell type, $5–10 \times 10^6$ cells were harvested for mRNA or protein isolation. After two rounds of washing with 1X PBS, cell pellets were lysed and treated accordingly the isolation purposes. RNA was isolated using the RNeasy Plus kit (Qiagen). For protein isolation, cells were resuspended in RIPA lysis buffer, vortexed thoroughly, and incubated on ice for 1 h. Afterwards, the content was harvested, and the protein cell lysate supernatant was stored to be used for SDS-PAGE.

## mRNA expression by qRT-PCR

Total RNA was isolated from PBMC, CD14+ isolated cells, undifferentiated BLaER1 and transdifferentiated BLaER1, and THP-1 cells as described above. Next, cDNA was synthesized using the RevertAid First Strand cDNA Synthesis Kit (Thermo Fisher Scientific). To detect *RNASE6*, *RNASET2*, and *RNASE2* expression, cDNA was amplified using the SYBR™ Green PCR Master Mix (Applied Biosystems). Primer sequences are described in Appendix Table S2. Fold change in relative gene expression was normalized based on β2-Microglobulin housekeeping expression.

## Detection of RNase proteins by mass spectrometry

For the identification of RNase proteins in various cell lines, THP-1 and transdifferentiated BLaER1 cells were harvested and washed in 1X DPBS (Gibco) twice, before lysis in 1.125% SDS (Carl Roth) heated at 99 °C for 10 min. The proteome samples were processed using an in-gel digestion protocol essentially as described previously (Shevchenko et al, 2006; Scherer et al, 2020; Lüders et al, 2022). In brief, proteins denatured in 1X NuPAGE LDS sample buffer were separated in a 4–10% NuPAGE Bis-Tris gel using 1X NuPAGE MES SDS buffer (Thermo Fisher Scientific) and stained with Coomassie G250 (Carl Roth). Following reduction with 10 mM dithiothreitol (Merck) and alkylation with 50 mM iodoacetamide (Merck), proteins were digested in 50 mM ammonium bicarbonate buffer pH 8.0 with 1 µg MS-grade trypsin (Serva) per sample. Produced peptides were desalted and concentrated using StageTips (3 M), before they were loaded on a nanoElute HPLC (Bruker) using a 25 cm Aurora ultimate column (Ionopticks). To separate peptides, a flow rate of 0.2 µl/min with a 100 min gradient from 0.1% formic acid to 95% acetonitrile/0.1% formic acid was used. Measurements were performed on a timsTOF Pro (Bruker) using data-dependent acquisition with a 1.1 sec cycle time. Other measurements were performed with an Easy-nLC 1200

coupled to an Orbitrap Exploris 480 mass spectrometer (Thermo Fisher). Resulting measurement files were analyzed using the standard settings in MaxQuant (2.1.0.0) with the human SwissProt and Trembl databases. Additionally, label-free quantification (LFQ) and intensity-based absolute quantification (iBAQ) were activated.

## Subcellular organelle fractionation

To determine the subcellular localization of specific proteins, a protocol published by Walker and colleagues (2016) for the isolation of endosomes was adopted. BLaER1 transdifferentiation was upscaled to 150 mm TC-treated culture dishes using the same growth conditions as described above. Following the harvest of $1.5–2.0 \times 10^8$ transdifferentiated cells in PBS, they were resuspended in RPMI for 60 min and kept at 4 °C for the remaining procedure. Afterward, cells were incubated for 10 min in 5 volumes of a hypoosmolar buffer made of 10 mM HEPES/KOH pH 7.6 (Carl Roth), 1.5 mM MgCl$_2$ (Sigma-Aldrich), and 10 mM KCl (Carl Roth). Cell homogenization was performed in 1 volume of 0.1% Igepal CA-630 (Sigma-Aldrich), 0.5 mM DTT (Merck), and 1X cOmplete Protease Inhibitor Cocktail (Roche) using a Dounce tissue grinder with 0.071 mm clearance (DKW Life Sciences). To reduce the amount of nuclei and large cellular fragments, the homogenate was centrifuged at $3300 \times g$ for 15 min, from which the post-nuclear supernatant (PNS) containing cellular organelles was recovered. Following the adjustment of the PNS to 150 mM NaCl (Sigma-Aldrich), 1 mM EDTA (Carl Roth), and 25% sucrose (Sigma-Aldrich), a discontinuous density gradient was prepared: 3 ml 45%, 4.1 ml 35%, and 3.8 ml 25% sucrose were layered from bottom to top in a 13.2 ml open-top thin wall ultra-clear tube (Beckman Coulter), the adjusted PNS was carefully layered on top. Ultracentrifugation was performed at $100,000 \times g$ and 4 °C for 1 h. Afterward, 2 ml fractions were collected from top to bottom and pooled from two gradients. For identification and quantification of proteins in each fraction, the in-gel digestion and HPLC-mass spectrometry measurement was performed as described above.

## Detection of RNase proteins by Western blotting

Cell lysate protein content was quantified by Pierce™ BCA Assay Kit (Thermo Scientific) for microplates and measured by Tecan microplate reader. For the SDS-PAGE, 5, 20, or 25 µg of protein was denatured in 4X SDS sample buffer (glycerol, β-mercaptoethanol, Tris-HCl pH 6.8, bromophenol blue, SDS) for 5 min at 95 °C. Then, samples were run in 4–20% gradient polyacrylamide precasted gels (anamed) the samples as well as a PageRuler™ Prestained Protein Ladder (Thermo Scientific). Electrophoresis was performed in PerfectBlue™ Vertical Double Gel System (peqlab) at 120 V for 90–120 min. Proteins were transferred to a PVDF membrane—preactivated in methanol for 5 min—for 1 h at 120 mA using a Biometra Fastblot™ system (Analytik Jena). Unspecific sites were blocked by incubation of the membrane in 1X Blue Block (SERVA) for 1 h under shaking at room temperature. Next, the membrane was cut to incubate it with different primary antibodies diluted in 1X Blue Block under shaking: rabbit anti-β-tubulin (1:1000, Cell Signaling Technology), mouse anti-β-actin (1:10,000, Cell Signaling), rabbit anti-RNase 6 (1:400, LifeSpan Biosciences), rabbit anti-RNase T2 (1:500, Sigma-Aldrich), rabbit anti-RNase 2 (1:500, Thermo Fisher), and goat anti-rabbit IgG secondary

antibody (1:500, Cell Signaling), or horse anti-mouse IgG (1:2000, Cell Signaling). Primary antibodies were incubated at 4 °C overnight under shaking. Thereafter, the membrane was washed in 1X TBS 0.1% Tween 20 for 5 min. This step was repeated two more times, and then the membrane was incubated with the conjugated secondary antibody for 1 h at room temperature under shaking. The detection of antibodies was developed by WESTAR ηC ULTRA 2.0 (Cyanagen), and membranes were finally imagined with Intas ChemoStar imager.

## Bacteria strains and culture conditions

The following bacteria strains were used for experiments: *Staphylococcus aureus* ATCC 25923, and *Enterococcus faecalis* ATCC 29212. Single colonies from blood agar plates were precultured in Luria-Bertani (LB) medium at 37 °C under agitation until reaching the mid-logarithmic phase. Thereafter, the bacteria were harvested at $400 \times g$ for 10 min. The pellets were processed per the intended following application.

## Total bacterial RNA extraction and isolation

Bacteria were lysed with 1 ml lysozyme (40 mg/ml in 10 mM Tris buffer, pH 7.5, SERVA). Then, the total bacterial RNA was extracted using TRIzol for phase separation and RNA precipitation, as suggested by the manufacturer. RNA purity was accessed by 260/230 and 260/280 absorbance measurements on a NanoDrop ND-1000 Spectrophotometer (Marshall Scientific), while RNA integrity was evaluated by running the samples on a 1.5% agarose gel which is detailed later.

## Serum-free whole live bacterial infection

To avoid interference from serum RNases, complete RPMI was replaced by RPMI free of antibiotics and serum before the infection. Isolated primary monocytes were pretreated with CU-CPT9a (10 µM)—a TLR8 inhibitor—1 h prior to infection. Next, regardless of the cell type, cells were infected with the respective indicated strains at different multiplicities of infection (MOI) for 1 hour or treated with TLR agonists. Then, gentamycin (50 µg/ml) was added on top of the wells followed by a 2-h incubation when cell were lysed for mRNA expression or a 20-h incubation when cytokine production was measured in the supernatant.

## Ex cellulo digestion assay

Reactions followed the protocol suggested by Greulich and colleagues (2019) with some adaptations. If not otherwise stated, 1 µg of bRNA or synthetic chimeric ORN was added to different concentrations of recombinant human RNase 6 (Creative Biomart) or RNase T2 (Cusabio) in Tris-EDTA buffer (pH 8.0, Sigma-Aldrich). After supplementing to 10 µl, the reaction samples were incubated for 20 min at 37 °C. Additionally, a loading control was prepared where RNases were not added. Afterward, 2X RNA Loading Dye (Thermo Fisher) was added in a 1:1 ratio to all samples and heated to 95 °C for 1 min.

Fragments out of bRNA digestion were visualized in 1.5% agarose gels prepared using diethyl pyrocarbonate (DEPC) water. For band visualization, peqGREEN (peqlab) was added during gel casting. Gels ran in 1X TAE buffer at 120 V for 1 h.

Synthetic ORNs fragments were analysed in 1X TBE urea denaturing polyacrylamide (20%, 29:1) gels. The urea gels were pre-run at 300 V for 30 min without samples using 1X TBE as a running buffer in a PerfectBlue™ Vertical Double Gel System (peqlab). After the pre-run, the diffused urea was washed away from the wells, and samples were run for 30 min at 300 V. Gels were washed for 10 min under shaking in 1X TBE and stained in 0.2% Methylene Blue staining solution for 1 h. Finally, gels were rinsed three times in distilled water for 10 min each step and subjected to an ethanol dehydration process previously described by Soto and Draper (2012). Briefly, the gel was transferred to ethanol 95% for 10 min followed by another incubation of 10 min and a final incubation of 5 min. Regardless of the gel type, all gels were visualized using the Intas ChemoStar imager.

### Stimulation of BLaER1 cells with bRNA or synthetic ORNs

bRNA was adjusted in Opti-MEM to the desired working concentration and incubated for 10 min at room temperature. Thereafter, the bRNA was complexed with poly-ʟ-arginine (pLArg) (Sigma-Aldrich) in Opti-MEM at a 1:1 (bRNA:pLArg) ratio, well mixed thoroughly by pipetting, and incubated for 20 min at room temperature. Finally, RPMI was added to complexes and mixed thoroughly again. Following the same rationale, single synthetic ORNs were complexed at a 1:1 ratio, while preparations with two different ORNs followed a 1:1:2 (ORN:ORN:pLArg) ratio. 50 µl of the RNA-pLArg complexes were added to the wells. Stimulation with 1 µg/ml of TL8-506 (InvivoGen) or LPS (InvivoGen) served as positive control. Plates were incubated for 20 h at 37 °C.

### Cytokine detection

Cytokines were measured in cell-free supernatant by BD OptEIA™ kits following the protocol suggested by the manufacturer. The absorbance at 450 nm was measured using a Tecan plate reader.

### Statistical analysis

Data were analyzed using GraphPad Prism 10. The applied statistical tests, number of experiments and replicates are described in each figure legend. $*p \leq 0.05$, $**p \leq 0.01$ $***p \leq 0.001$, and $****p \leq 0.0001$.

## Data availability

The mass spectrometry proteomics data have been deposited to the ProteomeXchange Consortium via the PRIDE (Perez-Riverol et al, 2021) partner repository with the dataset identifier PXD053881.

The source data of this paper are collected in the following database record: biostudies:S-SCDT-10_1038-S44319-024-00281-9.

## Peer review information

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

## Acknowledgements

We are thankful to Marie Eweleit and the technical support from Doreen Ussath on the optimization of RNase 6 detection by western blot. We also acknowledge technical support by Ina Ambiel on the generation of RNase 6 deficient human primary monocytes. This work was supported by the Deutsche Forschungsgemeinschaft (DFG, German Research Foundation)—project

number 439669440 TRR319, TP A03 and project number 246807620 TRR156, TP B05, both to AHD. OTF acknowledges supported by the Ministry of Science, Research and Culture Baden-Württemberg and the Deutsche Forschungsgemeinschaft (DFG, German Research Foundation) project number 259021520. Graphical abstract was created in BioRender: Valeriano Nunes, I. (2024) BioRender.com/c33m119.

## Author contributions

**Ivanéia V Nunes**: Conceptualization; Data curation; Formal analysis; Validation; Investigation; Visualization; Methodology; Writing—original draft; Writing—review and editing. **Luisa Breitenbach**: Conceptualization; Data curation; Formal analysis; Investigation; Visualization; Methodology; Writing—review and editing. **Sarah Pawusch**: Software; Formal analysis; Investigation; Visualization; Methodology; Writing—review and editing. **Tatjana Eigenbrod**: Conceptualization; Resources; Methodology; Writing—review and editing. **Swetha Ananth**: Formal analysis; Investigation; Methodology; Writing—review and editing. **Paulina Schad**: Formal analysis; Investigation; Methodology; Writing—review and editing. **Oliver T Fackler**: Supervision; Funding acquisition; Investigation; Methodology; Writing—review and editing. **Falk Butter**: Conceptualization; Data curation; Formal analysis; Supervision; Funding acquisition; Investigation; Project administration; Writing—review and editing. **Alexander H Dalpke**: Conceptualization; Formal analysis; Supervision; Funding acquisition; Investigation; Project administration; Writing—review and editing. **Lan-Sun Chen**: Conceptualization; Data curation; Formal analysis; Supervision; Validation; Investigation; Visualization; Methodology; Writing—original draft; Project administration; Writing—review and editing.

Source data underlying figure panels in this paper may have individual authorship assigned. Where available, figure panel/source data authorship is listed in the following database record: biostudies:S-SCDT-10_1038-S44319-024-00281-9.

## Funding

## Disclosure and competing interests statement

The authors declare no competing interests.

# Expanded View Figures

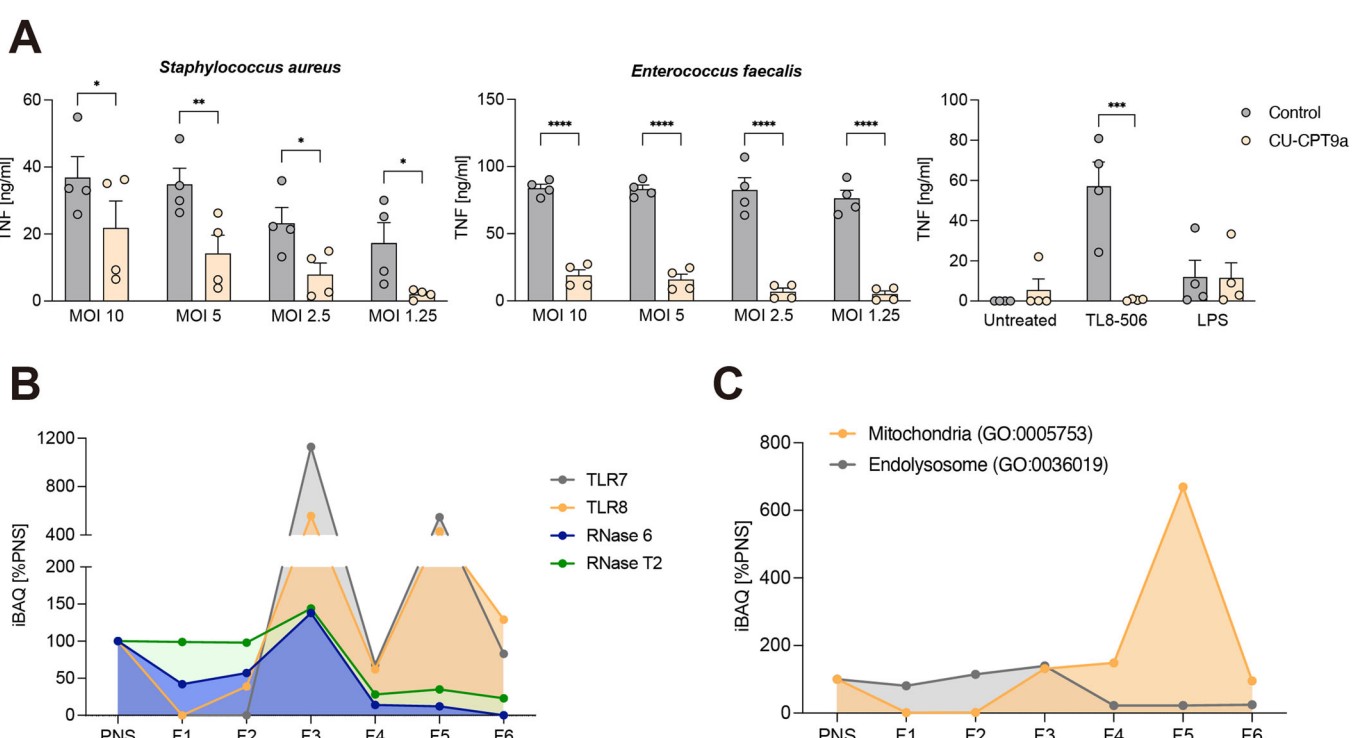

**Figure EV1. *RNASE6* is downregulated upon infection with TLR8-dependent bacteria.**

(A) Isolated monocytes were pre-treated with CU-CPT9a for 1 h, thereafter, cells were infected with different MOI. After 1 h, gentamycin was added to the wells. TNF-α was measured from the supernatant collected upon 20 h of incubation ($n = 2$, each with technical duplicates). Data were shown as mean ± SEM and analyzed by two-way ANOVA followed by Šídák's multiple comparisons test. Exact $p$ values for individual sub-panels are reported here from left to right: *Staphylococcus aureus*, *$p = 0.0343$, **$p = 0.0042$, *$p = 0.0314$, and *$p = 0.0309$; *Enterococcus faecalis*, ****$p < 0.0001$; Controls, ***$p = 0.0001$. (B) Post-nuclear supernatant (PNS) of transdifferentiated BLaER1 cells was fractionated by ultracentrifugation on a sucrose gradient. From lowest to highest density, fractions 1–6 (F1–F6) were pooled from two gradients for in-gel digestion and analysis by HPLC-mass spectrometry to determine intensity-based absolute quantification (iBAQ) values. RNase 2 values were below the limit of detection. (C) The distribution of organelles over the density gradient as in (B) is shown exemplarily for mitochondria and endolysosomes. In detail, the iBAQ values of all identified proteins belonging to the gene ontology term GO:0036019 Endolysosome were used as average and normalized to the PNS. For mitochondria, the same was applied to all proteins associated with GO:0005753 mitochondrial proton-transporting ATP synthase complex. Source data are available online for this figure.

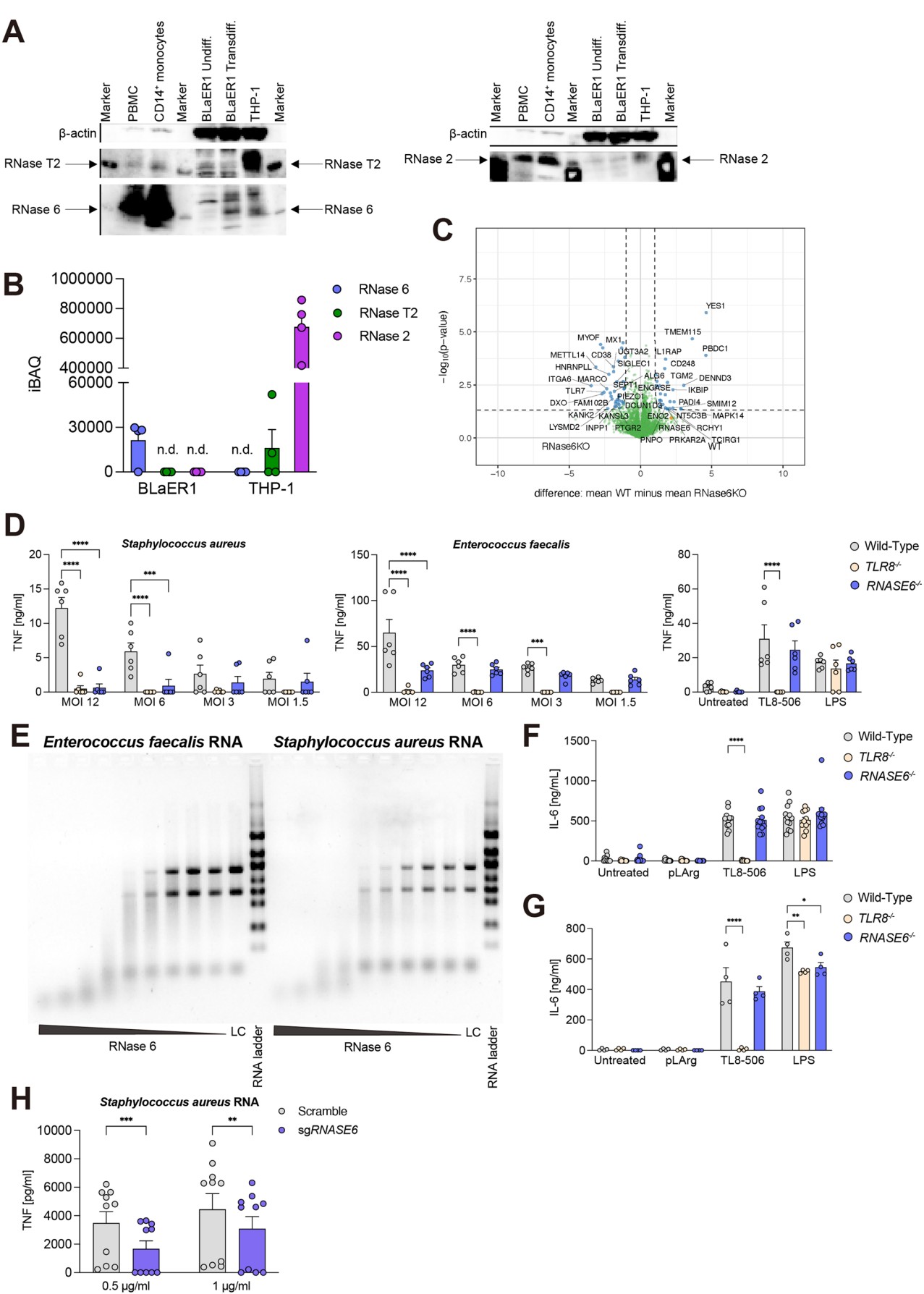

**Figure EV2.  RNase 6 is required to generate TLR8 active RNA fragments from bacterial RNA.**

(A) Immunoblot for RNase expression of the indicated cell types run with 25 µg of cell lysate. (B, C) Cellular proteomic analysis by HPLC-mass spectrometry to determine intensity-based absolute quantification (iBAQ) values comparing (B) BLaER1 and THP-1 cells or (C) Wild-type BLaER1 and *RNASE6*$^{-/-}$ cells from technical quadruplicates. The volcano plot was generated by R with the Welch *t*-test. The cutoff on the x-axis is for a twofold change and on the y-axis for a *p* value of 0.05. Measurements below the limit of detection are represented as not detected (n.d). (D) TNF-α detection by ELISA upon live whole bacterial infection with *S. aureus or E. faecalis* at several MOIs (*n* = 3, each with technical duplicates). Data were shown as mean ± SEM and analyzed with two-way ANOVA followed by Dunnett's test. Exact *p* values for individual sub-panels are reported here from left to right: *Staphylococcus aureus*, ****p* < 0.0001 and ***p* = 0.0006; *Enterococcus faecalis*, ****p* < 0.0001 and ***p* = 0.0002; Controls, ***p* = 0.0001. (E) 1 µg of isolated bRNA was incubated with different concentrations of recombinant human RNase 6 (10 to 0.025 ng/µl) for 20 min and visualized by agarose gel. An untreated sample was loaded as a control (LC, loading control). Representative image out of two gels. (F) Controls of experiments are shown in panel 2D in main Fig. 2 (*n* = 6, each with technical duplicates). Data were shown as mean ± SEM and analyzed with two-way ANOVA followed by Dunnett's test. The exact *p* value is ****p* < 0.0001. (G) Controls of predigested bRNA stimulation experiments are shown in panel 2E in main Fig. 2 (*n* = 2, each with technical duplicates). Data were shown as mean ± SEM and analyzed with two-way ANOVA followed by Dunnett's test. Exact *p* value are **p* = 0.0117, ***p* = 0.0022, and ****p* < 0.0001. (H) TNF-α detection upon *S. aureus* RNA stimulation in CD14+ monocytes edited by CRISPR-Cas9 (*n* = 2 in a total of five donors, each with technical duplicates). Data were shown as mean ± SEM and was analyzed with two-way ANOVA followed by Šídák's multiple comparisons test. The exact *p* values are ***p* = 0.0064 and ***p* = 0.0005. pLArg poly-ʟ-arginine, LPS lipopolysaccharide. Source data are available online for this figure.

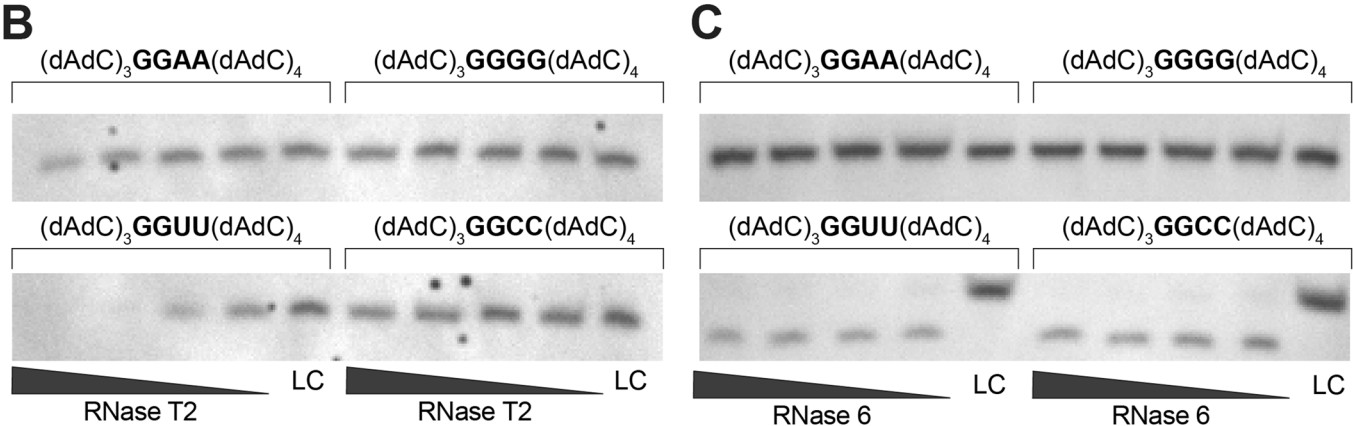

**Figure EV3. RNase 6 cleavage releases uridine-terminated RNA fragments.**

(A) Table of chimeric ORNs (protected deoxynucleotide sequences with central RNA) used for ex cellulo digestion by RNase 6 and RNase T2. (B) Urea gel of (dAdC)$_3$GGNN(dAdC)$_4$ digested by recombinant human RNase T2 or (C) RNase 6 (20 to 2.5 ng/µl). An untreated sample was loaded as a control (LC loading control). (B, C) Representative image out of two gels. N nucleotide, d deoxynucleotide, r ribonucleotide, A adenosine, G guanosine, U uridine, C cytosine). Source data are available online for this figure.

