## [Peer Review File · EMBO Reports]

Bacterial RNA sensing by TLR8 requires RNase 6 processing and is inhibited by RNA 2'O-methylation

Ivanéia Nunes, Luisa Breitenbach, Sarah Pawusch, Tatjana Eigenbrod, Swetha Ananth, Paulina Schad, Oliver Fackler, Falk Butter, Alexander Dalpke, and Lan-Sun Chen

Corresponding author(s): Alexander Dalpke (alexander.dalpke@med.uni-heidelberg.de)

Review Timeline:

Submission Date:	15th Mar 24
Editorial Decision:	15th Apr 24
Revision Received:	9th Aug 24
Editorial Decision:	3rd Sep 24
Revision Received:	17th Sep 24
Accepted:	19th Sep 24

Editor: Achim Breiling / Esther Schnapp

Transaction Report:

Dear Prof. Dalpke

Thank you for the submission of your manuscript to EMBO reports. I have now received the reports from the three referees that were asked to evaluate your study, which can be found at the end of this email.

As you will see, all referees have several comments, concerns, and suggestions, indicating that a major revision of the manuscript is necessary to allow publication of the study in EMBO reports. As the reports are below, and all the concerns need to be addressed, I will not detail them here.

Given the constructive referee comments, I would like to invite you to revise your manuscript with the understanding that the concerns of the referees must be addressed in the revised manuscript or in a detailed point-by-point response. Acceptance of your manuscript will depend on a positive outcome of a second round of review. It is EMBO reports policy to allow a single round of revision only and acceptance of the manuscript will therefore depend on the completeness of your responses included in the next, final version of the manuscript.

- 1) a .docx formatted version of the final manuscript text (including legends for main figures, EV figures and tables), but without the figures included. Figure legends should be compiled at the end of the manuscript text.
- 2) individual production quality figure files as .eps, .tif, .jpg (one file per figure), of main figures and EV figures. Please upload these as separate, individual files upon re-submission.

The Expanded View format, which will be displayed in the main HTML of the paper in a collapsible format, has replaced the Supplementary information. You can submit up to 5 images as Expanded View. Please follow the nomenclature Figure EV1, Figure EV2 etc. The figure legend for these should be included in the main manuscript document file in a section called Expanded View Figure Legends after the main Figure Legends section. Additional Supplementary material should be supplied as a single pdf file labelled Appendix. The Appendix should have page numbers and needs to include a table of content on the first page (with page numbers) and legends for all content. Please follow the nomenclature Appendix Figure Sx, Appendix Table Sx etc. throughout the text, and also label the figures and tables according to this nomenclature.

- 4) a complete author checklist, which you can download from our author guidelines (<https://www.embopress.org/page/journal/14693178/authorguide>). Please insert page numbers in the checklist to indicate where the requested information can be found in the manuscript. The completed author checklist will also be part of the RPF.

- 5) that primary datasets produced in this study (e.g. RNA-seq, ChIP-seq, structural and array data) are deposited in an

appropriate public database. If no primary datasets have been deposited, please also state this in a dedicated section (e.g. 'No primary datasets have been generated and deposited'), see below.

The accession numbers and database should be listed in a formal "Data Availability" section (placed after Materials & Methods) that follows the model below. This is now mandatory (like the COI statement). Please note that the Data Availability Section is restricted to new primary data that are part of this study. This section is mandatory. As indicated above, if no primary datasets have been deposited, please state this in this section

Data availability

7) Our journal encourages inclusion of *data citations in the reference list* to directly cite datasets that were re-used and obtained from public databases. Data citations in the article text are distinct from normal bibliographical citations and should directly link to the database records from which the data can be accessed. In the main text, data citations are formatted as follows: "Data ref: Smith et al, 2001" or "Data ref: NCBI Sequence Read Archive PRJNA342805, 2017". In the Reference list, data citations must be labelled with "[DATASET]". A data reference must provide the database name, accession number/identifiers and a resolvable link to the landing page from which the data can be accessed at the end of the reference. Further instructions are available at: <http://www.embopress.org/page/journal/14693178/authorguide#referencesformat>

8) Regarding data quantification and statistics, please make sure that the number "n" for how many independent experiments were performed, their nature (biological versus technical replicates), the bars and error bars (e.g. SEM, SD) and the test used to calculate p-values is indicated in the respective figure legends (also for EV figures and all those in an Appendix). Please also check that all the p-values are explained in the legend, and that these fit to those shown in the figure. Please provide statistical testing where applicable. Please avoid the phrase 'independent experiment', but clearly state if these were biological or technical replicates. Please also indicate (e.g. with n.s.) if testing was performed, but the differences are not significant. In case n=2, please show the data as separate datapoints without error bars and statistics. See also: <http://www.embopress.org/page/journal/14693178/authorguide#statisticalanalysis>

9) Please add scale bars of similar style and thickness to microscopic images, using clearly visible black or white bars (depending on the background). Please place these in the lower right corner of the images themselves. Please do not write on or near the bars in the image but define the size in the respective figure legend.

10) Please also note our reference format:

12) We now use CRedit to specify the contributions of each author in the journal submission system. CRedit replaces the author contribution section. Please use the free text box to provide more detailed descriptions and do not provide your final manuscript text file with an author contributions section. See also our guide to authors: <https://www.embopress.org/page/journal/14693178/authorguide#authorshipguidelines>

13) We would encourage you to use 'Structured Methods', our new Materials and Methods format. According to this format, the

Materials and Methods section should include a Reagents and Tools Table (listing key reagents, experimental models, software, and relevant equipment and including their sources and relevant identifiers), uploaded as separate file, followed by a Methods and Protocols section in which we encourage the authors to describe their methods using a step-by-step protocol format with bullet points, to facilitate the adoption of the methodologies across labs. More information on how to adhere to this format as well as downloadable templates (.doc or .xls) for the Reagents and Tools Table can be found in our author guidelines (section 'Structured Methods'):

14) Please add up to five keywords to the manuscript, remove the summary and provide the abstract written in present tense.

Please also order the manuscript sections like this, using these names:

Title page - Abstract - Keywords - Introduction - Results & Discussion - Methods - Data availability section - Acknowledgements - Disclosure and Competing Interests Statement - References - Figure legends - Expanded View Figure legends

I look forward to seeing a revised version of your manuscript when it is ready. Please let me know if you have questions or comments regarding the revision.

Yours sincerely,

Referee #1:

Summary

1. Does this manuscript report a single key finding?

YES, the manuscript single key finding is that RNase 6 contributes in degradation of bacterial RNA in a monocytic BlaER cells, and thus generating TLR8-ligands for TLR8-dependent sensing of bacterial RNA and entire bacteria.

2. Is the reported work of significance (YES), or does it describe a confirmatory finding or one that has already been documented using other methods or in other organisms etc (NO)?

YES, as far as I am concerned, it has not shown previously that RNase 6 indeed contribute to formation of TLR8-activating degradation products, although it has been suggested.

3. Is it of general interest to the molecular biology community?

YES, the paper does at least to some extent explain molecular requirements for processing of RNA by RNase 6, which is linked to the important biological effect they revealed.

4. Is the single major finding robustly documented using independent lines of experimental evidence (YES), or is it really just a preliminary report requiring significant further data to become convincing, and thus more suited to a longer-format article (NO)?

YES, the authors provide evidence in the form of a KO-cell line, as well with a rescue experiment. Still, it could be possible to further strengthen the evidence of this suggested important role of RNase 6 by including some studies on primary human cells.

Report:

TLR8 is a sensor RNA within myeloid cells and has a role in the detection pathogens, including entire bacteria. Here, Nunes et al. claim that RNase 6 has a role in the processing of bacterial RNA, resulting TLR8-dependent cytokine production and innate immune sensing of bacteria, including live *S. aureus* and *E. faecalis*.

TLR8 is activated by the RNA degradation products uridine (ribomononucleoside), binding to TLR8-site 1, and uridine-containing 2- or 3-mers, binding to site 2. Recent studies by Greulich et al. and Ostendorf et al. revealed how RNase T2 and RNase 2 are critical for formation of these TLR8-activating ligands from bacterial RNA, live pathogens, and synthetic oligomers. Greulich et al. also examined the effect of RNase 6 in BlaER cells, since it was highly expressed, but failed to reveal effect on TLR8-activation by an RNA oligomer.

On this background, the current findings by Nunes et al. on RNase 6 are somewhat surprising, even though another recent study recently revealed how RNase 6 contribute to TLR7 activation in pDCs, and that it likely can contribute to formation of TLR8 ligands.

Detailed considerations:

1. That RNase 6 has a role in TLR8 activation by bacteria is a significant claim, and a novel and somewhat unexpected finding considering the background knowledge (Greulich et al)

2. The claim is based on convincing data, although limited by a single KO cell line, it was also complemented with a rescue experiment.
3. The claims are largely discussed in a proper manner in the context of previous studies, though some improvements are suggested below.
4. The claim is likely to be of interest to the biological/immunological scientific community, since it is relevant for a central mechanism in host-pathogen interactions and may also extend to several other inflammatory diseases and conditions where TLR8 could play a role (e.g. cancer, autoimmunity, allergy and vaccination).
5. From a technical point of view, the study does not stand out from others in its field. However, it has a clear message with potential impact for a relative broad research field, as stated above.
6. This main message is well supported but could be improved by also including some studies on primary human cells.

Suggested improvements on writing and presentation of data:

1. The statement that TLR8 induces MyD88-dependent signaling with NF- κ B and IRF3 activation lacks reference, and newer studies rather pointing towards IRF5 are not mentioned.
2. It is unclear if the authors suggest that TLR8 site 1 also bind ssRNA fragments, or only monouridines. Binding of ssRNA fragment to both sites are also shown in the graphical abstract. Are there actually any evidence for this, or could it be made more clear that site 1 only binds monouridine?
3. The sentence claiming that addition of a 2'O-Me at the uridine UA converts the sequence to the RNase 6 preferential cleavage, seems erroneous, opposite of what is shown?
4. The claim that the findings strengthen the physiological role of TLR8 for sensing a range of bacteria seems to oversell the story, since TLR8-dependent sensing of *E. faecalis* and *S. aureus* has been done previously.
5. It is possible that transcriptional downregulation of RNases upon TLR8 activation could indicate counterregulation. However, it might not be a specific effect, but rather represent metabolic changes or induction of cell death.
6. RNase 2 data in Figure 1D is not shown, although the text indicate it is.
7. Figure 2A: Subfigures are normalized to PBMC (1.0)? The relative expression of each RNase between cells lines are thus not given, and it is difficult to follow the discussion of the relative levels. Suggest representing the fold versus b2M only, on a common y-axis (and not calculate as fold vs PBMCs). Moreover, RNase expression in these cell types are previously examined by RNAseq by Greulich et al, Ostendorf et al., and Tong et al., who show RNase-T2, -2, and -6 are well expressed in classical monocytes, while differentiated BLAER1 express RNASE-T2, -1 and -6, but not -2. Thus, except for higher RNase 2 in BLAER1, Fig 2A seems mostly in accordance with these earlier findings.
8. Figure 2C: Seems a strong effect for *S. aureus*, but little effect for the enterococcus. Thus, suggest not to oversell the interpretation in the text (profoundly...)
9. RNase 6-dependent IL-6 with bRNA is shown in 2E (not 2C).
10. Tonge et al. added 2'O-Me at the first uridine (U1) in pUUC (not the second, as stated), corresponding to UmUAA in this manuscript, and it was degraded by RNase 6. In contrast, their UmC sequence (pUC-U2'O) was protected. Altogether, however, this seems in agreement with the claims.

Suggested additional experiments:

1. The study is based on a single monocytic cell line (BLAER1). It would be interesting to know if RNase 6 has a similar role in primary myeloid cells, such as primary human monocytes or macrophages. This might be examined by siRNA, or alternative by a KO strategy like that used by Tong et al?
2. Moreover, the relative impact of RNase 6, RNase T2 and RNase 2 for sensing of bacteria in primary myeloid cells was not examined. Clarifying this issue would improve the current study. Parts of Fig. 1 might be moved to supplementary if additional data are provided.

Referee #2:

The manuscript by Nunes et al. reports that bacterial RNA is sensed by TLR8 in a RNase6 dependent manner. This work is an extension of previous work by the Hornung (Greulich et al) and Bartok group (Ostendorf et al.) that demonstrated that RNase T2 and RNase 2 are important in the generation of RNA degradation products from synthetic ORN (and bacterial RNA from *Staphylococcus aureus*). , this submitted work also extends studies on the role of 2'O-methylation of RNA on the inhibition of RNase activity and TLR activation potential (as reported by Tong et al.).

Figure 1 demonstrates the bacterial induced cytokine production in in vitro infections assays using WT and TLR8 deficient transdifferentiated BLAER1 cells is TLR8 dependent. The data for the TLR8 deficient clone used looks convincing. However, to rely on a single clone may be problematic. The use of an alternative system such as a TLR8 inhibitor on WT BLAER1 cells (or primary human monocytes) could strengthen the authors claim.

Interestingly, Figure 2 demonstrates that that in vitro infection with *S. aureus* is TLR8 and RNase 6 dependent, whereas *E.*

faecalis is only TLR8 dependent. In contrast, isolated RNA from both strains induces IL-6 in a TLR8 and RNase6 dependent manner. Are the results shown here consistent with all three RNase6 deficient clones (Fig. 2B). RNase6 degraded RNA also can reconstitute immunostimulation in RNase6 deficient cells similar to T2 RNase fragments reported by Greulich et al.

Figure 3 clearly demonstrates that RNase6 creates U-terminated RNA fragments in a synthetic construct that is not cleaved by recombinant RNase T2. Although latter is expected a positive control for RNase T2 activity would informative. This sequence should not be cut by RNase6.

Figure 4 investigates the role of 2'O-methylation on RNA cleavage by RNase6. This work extends published work on RNase T2 and RNase6 by demonstrating that UA is the preferential cleavage site that can be inhibited by U -2'-O methylation (Um).

In general, the discussion avoids to challenge the role of RNase T2 and RNase 2 in the generation of RNA degradation products for TLR8 activation. Following these original publications, RNase 6 should not play a prominent role for TLR8 activation. The authors could speculate more about this discrepancy. Is it possible that RNase6 knockout has an influence on the expression of RNase T2 /RNase 2 and vice versa? The RNase6 deficient clones probably could be tested for the other RNases by western blotting.

Overall the manuscript is important since it introduces a new player in the field of RNA degradation dependent TLR8 activation.

Referee #3:

The present work "Bacterial RNA sensing by TLR8 requires RNase 6 processing and is inhibited by RNA 2'O-methylation" deals with the immune recognition of viral RNA by TLR8 on the example of the two bacteria *S. Aureus* and *E. faecalis*. In particular, the importance of RNase-dependent RNA degradation is discussed, as it has recently been shown that individual methylation positions influence RNA degradation and thus also immune recognition by TLR8. It is a well-written paper and an interesting finding. Most of my comments relate to minor ambiguities or missing citations. However, I do not have the impression that the RNase 6 dependence of *E. faecalis* detection is sufficiently supported by the data, so I would like to ask for further experiments and clarifications here.

I would like to request the following revisions:

Abstract:

"Herein, we characterized molecular RNase 6 cleavage mechanisms. BLaER1 RNASE6^{-/-} cells showed a dampened TLR8-dependent response upon stimulation with isolated bacterial RNA (bRNA) but also upon infection with live whole bacteria." Perhaps write "and" instead of "but"? I first wondered where the contradiction should be.

Introduction:

Please at citations after the following sentences:

- Small synthetic agonists, such as the imidazoquinolinone derivat R848 and the benzodiazepine compound TL8-506, can efficiently activate the receptor by solely binding to the apex site.
- Regarding their cleavage selectivity, it was shown that RNase T2 preferentially cleaves in between purines and uridine, and RNase A family members in general selectively cleave in between pyrimidines and purines.
- Among the several RNase A family members, RNase 2 was the first one to be shown to process ssRNA in the endosome.
- RNase 2 expression is mostly upregulated in eosinophils and basophils while in monocytes low expression has been observed.

Please clarify this point:

"RNase 6 exhibits extracellular antimicrobial peptide (AMP) activity against a range of bacteria including *Escherichia coli* and *Staphylococcus aureus* by the induction of bacterial membrane depolarization and bacterial agglutination at low micromolar concentration (Pulido et al., 2016). Interestingly, RNase 6 upregulation upon urinary tract infection (UTI) with *E. coli* has been reported and RNASE6 transgenic mice showed increased protection upon experimental UTI with uropathogenic *E. coli* (Becknell et al., 2015; Ruiz-Rosado et al., 2023)."

Could you please interlink this part a bit better? When reading it, it is not directly clear why. Also, it seems contradictory to the first headline in the results part; please discuss / clarify this.

Results and discussion:

"We measured proinflammatory cytokines in the cell-free supernatant of BLaER1 cells lacking RNASE6 upon infection with bacteria."

Could you rearrange this sentence? I know what you want to say, but at first reading, it sounds like these cells lack RNase 6 when they are infected.

Figure 2c and 2E:

Is *E. faecalis* sensing really depending on RNase 6 digestion? From these pictures, it does not look like this to me. The decrease in RNase 6 ko cells is really minimal, and in Figure 2E, IL-6 induction is even stronger in RNase 6 deficient cells. To keep these data in the paper, it would be essential to clarify this point, perhaps by RNase T2 ko and RNase 6 and T2 double ko.

Figure 2D:

The two blots are impressively similar, so I checked several times to make sure that the same blot was not accidentally shown twice with different exposures. However, due to differences in the ladder intensity, I am now convinced that two different blots have been shown and that the work was just impressively accurate. Nevertheless, to be on the safe side, I would like to ask for this to be double-checked in-house.

Figure 2E: Please discuss this figure in the results / discussion section.

Most important point: Please clarify RNase 6 and / or RNase T2 dependency of *E. faecalis* and discuss this difference in more detail.

Point-by-point reply, EMBOR-2024-59209V1

We thank the reviewers for their careful consideration of our manuscript. Based on their helpful comments we have now revised our manuscript including additional as well as entirely new experimental data. The revised manuscript addresses all points raised by the reviewers. We hope that the revised manuscript is now acceptable for publication.

Referee #1:

Suggested improvements on writing and presentation of data:

1. The statement that TLR8 induces MyD88-dependent signaling with NF- κ B and IRF3 activation lacks reference, and newer studies rather pointing towards IRF5 are not mentioned.

We have revised the manuscript accordingly, considering that indeed IRF5 plays an important role in TLR8 activation (p. 3, l. 11). References (Bergstrøm *et al*, 2015; Heinz *et al*, 2020) were added.

2. It is unclear if the authors suggest that TLR8 site 1 also bind ssRNA fragments, or only monouridines. Binding of ssRNA fragment to both sites are also shown in the graphical abstract. Are there actually any evidence for this, or could it be made more clear that site 1 only binds monouridine?

In fact, we did not address the question whether site 1 may bind ssRNA fragments. As pointed out by the reviewer, evidence so far indicates that site 1 binds monouridine. We therefore modified the graphical abstract as well as the text (p. 3, l. 16) to avoid false interpretations and misunderstandings.

3. The sentence claiming that addition of a 2'O-Me at the uridine UA converts the sequence to the RNase 6 preferential cleavage, seems erroneous, opposite of what is shown?

Thanks for pointing out this mistake. We now changed the sentence to "The addition of a 2'O-Me at the uridine in the dinucleotide ,UA' - an RNase 6 cleavage site - prevented cleavage and, therefore, the generation of uridine-terminated fragments." (p. 4, l. 30)

4. The claim that the findings strengthen the physiological role of TLR8 for sensing a range of bacteria seems to oversell the story, since TLR8-dependent sensing of E. faecalis and S. aureus has been done previously.

Following the suggestion, we have now reworded the sentence: "Therefore, our findings corroborate with previous work." (p. 5, l. 25)

5. It is possible that transcriptional downregulation of RNases upon TLR8 activation could indicate counterregulation. However, it might not be a specific effect, but rather represent metabolic changes or induction of cell death.

We modified the sentence to: "a possible mechanism of counterregulation. Yet, we cannot exclude other mechanisms." (p. 5, l. 31) From our experiments with stimulation for 3h we had no indications that cell death occurred (at least the overall RNA amount extracted was comparable in all situations and we observed good induction of proinflammatory genes *IL6* and *CXCL10* in infected conditions (Figure 1C lower row)). Yet we cannot exclude that e.g. metabolic changes contribute to the transcriptional downregulation.

6. RNase 2 data in Figure 1D is not shown, although the text indicate it is.

We re-analysed the mass spectrometry data from the fractionation experiment, and could not detect RNase 2 from all post-nuclear fractions in transdifferentiated BLaER1 cells (Figure EV 1B). It is probable that RNase 2 is less abundant in BLaER1 cells, whereas more abundant in THP-1 and primary CD14⁺ cells (Figure EV 2A and 2B). Of note, high expression of RNase 2 seems to be a peculiarity of THP-1 cells (Fig. 2A). We revised the text to: "This analysis revealed that RNase 6 and RNase T2 were detected in the

same cell fraction as TLR8 and TLR7, whereas RNase 2 expression was below the threshold of detection (Figure EV 1B).” (p. 6, l. 7)

7. Figure 2A: Subfigures are normalized to PBMC (1.0)? The relative expression of each RNase between cells lines are thus not given, and it is difficult to follow the discussion of the relative levels. Suggest representing the fold versus b2M only, on a common y-axis (and not calculate as fold vs PBMCs). Moreover, RNase expression in these cell types are previously examined by RNAseq by Greulich et al, Ostendorf et al., and Tong et al., who show RNase-T2, -2, and -6 are well expressed in classical monocytes, while differentiated BLaER1 express RNASE-T2, -1 and -6, but not -2. Thus, except for higher RNase 2 in BLAER1, Fig 2A seems mostly in accordance with these earlier findings.

Previously, the data were shown with $\Delta\Delta Ct$ method (Livak & Schmittgen, 2001), in which we normalized to PBMC. To allow assessment of relative expression levels subpanel 2A has now been changed according to the reviewer’s suggestions (Figure 2A). With regard to the protein expression, we have now added new experiments analysing total cell lysates from different cell types by immunoblotting for RNase 2, 6, and T2 (new Figure EV 2A). Western blot data matched our qPCR finding, which are in accordance with previous publications (Greulich *et al*, 2019; Ostendorf *et al*, 2020; Tong *et al*, 2023).

Combining the new results, we have re-structured the content in the manuscript: “We next assessed the relative *RNASE6*, *RNASET2*, and *RNASE2* mRNA and protein expression in the human monocytic THP-1 cell line as well as BLaER1 cells at B-cell undifferentiated and monocyte-like transdifferentiated stages. The mRNA expression levels were compared to peripheral blood mononuclear cells (PBMCs) and CD14⁺ isolated monocytes. All RNases showed mRNA expression in primary cells, but none or low expression in undifferentiated BLaER1 cells (Figure 2A). *RNASE6* and *RNASET2* expression was upregulated in transdifferentiated BLaER1 cells which showed low *RNASE2* expression. On the other hand, THP-1 cells highly expressed *RNASE2* and *RNASET2* whereas *RNASE6* could not be detected to that extent (Figure 2A).

The detection of RNases at the protein level by western blot revealed similar findings for the cell lines (Figure EV 2A). CD14⁺ monocytes highly expressed RNase 6, followed by RNase 2 to a lesser extent while RNase T2 showed the lowest expression (Figure EV 2A). Protein detection by western blot showed RNase 6 detection in BLaER1 cells and THP-1. Moreover, mass spectrometry confirmed RNase 6 detection in BLaER1 cells (Figure EV 2B). Overall, these findings follow those observed in previous studies (Ostendorf *et al*, 2020; Tong *et al*, 2023), arguing for RNase 6 being more important for RNA processing upstream of TLR8 activation since CD14⁺ monocytes are characterized as highly RNase 6 expressing cells and this was best mirrored in transdifferentiated BLaER1 cells. Based on these findings, we genomically edited BLaER1 cells by CRISPR-Cas9 to lack RNase 6 which was confirmed by antibody detection and mass spectrometry (Figure 2B, EV 2C).” (p. 6, l. 17)

8. Figure 2C: Seems a strong effect for *S. aureus*, but little effect for the enterococcus. Thus, suggest not to oversell the interpretation in the text (profoundly...)

Following the suggestion, we revised the manuscript to avoid overselling: “Live whole bacterial infection with *S. aureus* in *RNASE6*^{-/-} cells showed a strong and significant reduction of IL-6 and TNF- α secretion (Figure 2C, EV 2D). Despite the strong dependency on TLR8 recognition (Figure 1B), live infection with *E. faecalis* in RNase 6 deficient cells only resulted in a partial, yet significant reduction of cytokine secretion at highest MOI.” (p. 7, l. 10)

9. RNase 6-dependent IL-6 with bRNA is shown in 2E (not 2C).

We apologize for this mistake. In the revised manuscript the panel is now shown in Figure 2D.

10. Tonge et al. added 2'O-Me at the first uridine (U1) in pUUC (not the second, as stated),

corresponding to UmUAA in this manuscript, and it was degraded by RNase 6. In contrast, their UmC sequence (pUC-U2'O) was protected. Altogether, however, this seems in agreement with the claims.

We have corrected our manuscript focusing on UmC: “Accordingly, the addition of 2'O-Me at the uridine in pUC also drastically affected the generation of fragments as the modification prevented RNase 6 cleavage (Tong *et al*, 2023).” (p. 11, l. 25) Indeed, as identified by the reviewer, our findings with 2'O-Me positioning are in agreement with findings by Tong for pUmUC and UmC sequence (pUC-U2'O).

Suggested additional experiments:

1. The study is based on a single monocytic cell line (BlaER1). It would be interesting to know if RNase 6 has a similar role in primary myeloid cells, such as primary human monocytes or macrophages. This might be examined by siRNA, or alternative by a KO strategy like that used by Tong et al?

This is an important concern, also raised by Referee #3. We therefore undertook considerable efforts and performed gene editing of *RNASE6* in primary human CD14⁺ monocytes (hCD14⁺) via CRISPR-Cas9 using nucleofection. The new data in primary cells confirm our findings in BlaER1 cells. The new data from hCD14⁺ cells are now depicted in new Figure 2F-I and EV 2H. As shown in Figure 2G, sensing of *E. faecalis* RNA but also of bRNA *S. aureus* was clearly dependent on RNase 6 in primary human monocytes. Furthermore, pre-exposure of *E. faecalis* bRNA with recombinant RNase 6 exacerbated TNF secretion in both Scramble and sg*RNASE6* cells (Figure 2I). These new data from primary hCD14⁺ confirm our finding in BlaER1 cells, indicating an important role of RNase 6 in ssRNA sensing, specifically also in monocytes. We think that this new data is of high relevance and answers the question of the reviewers. Results are added as new paragraph on p. 8. l. 19-32 and Figure 2F-I and EV 2H.

2. Moreover, the relative impact of RNase 6, RNase T2 and RNase 2 for sensing of bacteria in primary myeloid cells was not examined. Clarifying this issue would improve the current study. Parts of Fig. 1 might be moved to supplementary if additional data are provided.

We agree that it will be interesting to study the relative impact of the different RNases for sensing of bacteria. However, generation of double and triple knockouts in BlaER1 cells or – using the newly established protocol for editing of primary cells (See above) – in monocytes is beyond the possibilities and time frame for this revision. Based on the new results showing that lack of RNase 6 in primary monocytes reduces or even abolishes bRNA sensing we think that we have now unequivocally identified RNase 6 as an additional player upstream of TLR8. Further experiments with respect to the relative contributions of different RNases to bacterial sensing must be subject of a follow-up study.

Referee #2:

Figure 1 demonstrates the bacterial induced cytokine production in in vitro infections assays using WT and TLR8 deficient transdifferentiated BLaER1 cells is TLR8 dependent. The data for the TLR8 deficient clone used looks convincing. However, to rely on a single clone may be problematic. The use of an alternative system such as a TLR8 inhibitor on WT BLaER1 cells (or primary human monocytes) could strengthen the authors claim.

We used TLR8 deficient BLaER1 cells to make the point that the model bacteria we are using are indeed sensed via TLR8, which is a prerequisite for RNase experiments. We did not intend to focus on TLR8 dependency in more detail. However, as suggested by the reviewer (and also based on the comments to verify results in primary cells) we now performed live infection with 2 different bacteria in primary human PBMC, pre-treated with TLR8 inhibitor (CU-CPT9a, 10 μ M). The results are now shown in Figure EV 1A, demonstrating the sensing of both *S. aureus* and *E. faecalis* is highly TLR8-dependent in primary PBMC. We also added on p. 5, l. 13: "We confirmed TLR8 dependency in primary human monocytes using CU-CPT9a, a TLR8 antagonist (Figure EV 1A)."

Interestingly, Figure 2 demonstrates that that in vitro infection with *S. aureus* is TLR8 and RNase 6 dependent, whereas *E. faecalis* is only TLR8 dependent. In contrast, isolated RNA from both strains induces IL-6 in a TLR8 and RNase6 dependent manner. Are the results shown here consistent with all three RNase6 deficient clones (Fig. 2B). RNase6 degraded RNA also can reconstitute immunostimulation in RNase6 deficient cells similar to T2 RNase fragments reported by Greulich et al. Indeed we observe strong RNase 6 dependency for infection with *S. aureus* in BLaER1 cells, whereas for *E. faecalis* partial reduction is only observed at high MOIs. Following the suggestions, we now restructured the sentences to: "Live whole bacterial infection with *S. aureus* in *RNASE6*^{-/-} cells showed a strong and significant reduction of IL-6 and TNF- α secretion (Figure 2C, EV 2D). Despite the strong dependency on TLR8 recognition (Figure 1B), live infection with *E. faecalis* in RNase 6 deficient cells only resulted in a partial, yet significant reduction of cytokine secretion at highest MOI." (p. 7, l. 10)

We had produced additional clones and based on the suggestion of the reviewer we now include new data from another independent clone of *RNASE6*^{-/-} BLaER1 cells (clone #78G9) below. Transfecting 2 μ g/ml of *E. faecalis* and *S. epidermidis* bRNA to transdifferentiated BLaER1 cells demonstrates significant reduction of IL-6 secretion in *RNASE6*^{-/-} #78G9 clone, which reinforce the phenotype in Figure 2D. Since this clone was only tested initially, we haven't conducted results from this in further experiments.

Beside of additional clones from BLaER1 cells, we have followed the supportive suggestions from the other reviewers, and have additionally performed gene editing of *RNASE6* in primary human CD14⁺ monocytes (hCD14⁺) via CRISPR-Cas9 to confirm findings in the BLaER1 cells. The new data from hCD14⁺ cells are now depicted in new Figure 2F-I and EV 2H. As shown in Figure 2G, sensing of *E. faecalis* bRNA but also of *S. aureus* bRNA was clearly dependent on RNase 6 in primary human monocytes. Furthermore, pre-exposure of *E. faecalis* bRNA with recombinant RNase 6 exacerbated TNF- α secretion in both Scramble and sg*RNASE6* cells (Figure 2I). These new data from primary hCD14⁺ confirm our finding in BLaER1 cells, indicating an import role of RNase 6 in ssRNA sensing, specifically also in monocytes. We think that this new data is of high relevance and answers the question of the reviewers.

Figure 3 clearly demonstrates that RNase6 creates U-terminated RNA fragments in a synthetic construct that is not cleaved by recombinant RNase T2. Although latter is expected a positive control for RNase T2 activity would be informative. This sequence should not be cut by RNase6.

To further assess the cleavage specificity of RNase 6 and RNase T2, we designed GGNN also flanked by non-stimulatory deoxynucleotide sequences. As predicted, RNase T2 was able to cleave only GGUU (Figure EV 3B) which is consistent with previous data showing preferential RNase T2 cleavage between GU releasing guanosine-terminated fragments for TLR8 concave site (Greulich *et al*, 2019).

Now we've reformatted the sentences to: "Upon cleavage by RNase 6, GGUU and GGCC sequences were degraded (Figure EV 3C). RNase A family members and RNase T2 show contrasting base preferences for their site of cleavage. Still, we observed degradation of GGUU by both RNase T2 and RNase 6. Therefore, we focused on deciphering where RNase 6 could cleave on GGUU. We noticed that there was a rUdA site on the (dAdC)₃GGUU(dAdC)₄ sequence which could perhaps serve as a site cleavage for RNase 6. Therefore, we engineered the (dAdC)₃U(dAdC)₄ sequence (Figure 3A) which was degraded by RNase 6 as well, but not by RNase T2 (Figure 3E). Similarly, a fluorescent substrate 6-FAM~dArUdAdA~6-TAMR showed cleavage by RNase 7, another RNase A family member (Rademacher *et al*, 2019). These findings suggest that RNase 6 is able to cleave when uridine is positioned at the B1 site and followed by deoxy adenosine in chimeric sequences also arguing that the cleavage on (dAdC)₃UUUU(dAdC)₄ and (dAdC)₃GGUU(dAdC)₄ might derive from the provided rUdA." (p. 9, l. 29) During the limiting timeframe of revision, we could not pin down where was the site cleavage in these sequences containing rUdA.

Regarding to collaboration between RNases, we have added further sentences of discussion: "Based on previous data, it is unlikely that RNase 6 and RNase T2 are preferentially cleaving at the same position in those sequences (GU or GC). A similar approach could be applied to pin down the site cleavage in GGCC. Of note, RNase 6 has shown catalytic activity on CpA (Prats-Ejarque *et al*, 2016). However, C-terminated sequences have not shown potential effect on TLR8 stimulation and, therefore, this sequence was not further analysed in this study. Taken together, these findings highlight that RNase 6 generates uridine-terminated fragments and in collaboration with RNase T2 – which releases guanosine-terminated products – can activate TLR8." (p. 10, l. 9)

Figure 4 investigates the role of 2'O-methylation on RNA cleavage by RNase6. This work extends published work on RNase T2 and RNase6 by demonstrating that UA is the preferential cleavage site that can be inhibited by U -2'-O methylation (Um).

In general, the discussion avoids to challenge the role of RNase T2 and RNase 2 in the generation of RNA degradation products for TLR8 activation. Following these original publications, RNase 6 should not play a prominent role for TLR8 activation. The authors could speculate more about this discrepancy. Is it possible that RNase6 knockout has an influence on the expression of RNase T2 /RNase 2 and vice versa? The RNase6 deficient clones probably could be tested for the other RNases by western blotting.

We appreciate your comments and suggestions. RNase 6 deficient BLaER1 clones have been tested for at least RNase T2 production and showed no difference when compared to Wild-Type cells. The original blot from Figure 2B is now depicted in the respective source data for that panel (see below). In addition to Western Blot, we have now conducted total proteome analysis from Wild-Type and RNase 6 deficient BLaER1 cells for this revision (Figure EV 2C). From analysis on significant enriched proteins, we didn't see the other RNases were altered in RNase 6 deficient cells. Intriguingly, we found TLR7 and MX1 proteins were enriched in RNase 6 deficient BLaER1 cells. This may suggest a compensatory effect due to one parts of ssRNA-sensing machinery (RNase 6 and TLR8) were altered, but we need to further characterize the cells in this regard.

To study the mentioned “discrepancy” we decided to analyze primary monocytes for RNase6 dependency of bRNA sensing: We therefore performed gene editing of *RNASE6* in primary human CD14⁺ monocytes (hCD14⁺). The new data in primary cells confirm our findings in BLaER1 cells (Figure 2F-I and EV 2H). As shown in Figure 2G, sensing of *S. aureus* bRNA but also of *E. faecalis* bRNA was clearly dependent on RNase 6 in primary human monocytes.

We agree that it would also be interesting to study the relative impact of the different RNases for sensing of bacteria. However, generation of double and triple knockouts in BLaER1 cells or – using the newly established protocol for editing of primary cells – in monocytes is beyond the possibilities and time frame for this revision. Based on the new results with primary monocytes (Figure 2F-I and EV 2H) showing that lack of RNase 6 in primary monocytes reduces or even abolishes bRNA sensing we think that we have now unequivocally identified RNase 6 as an additional player upstream of TLR8.

Point-by-point reply, EMBOR-2024-59209V1

Overall the manuscript is important since it introduces a new player in the field of RNA degradation dependent TLR8 activation.

Thanks for this positive and encouraging comment. With the added new experiments, we hope that we could reply to all the reviewer's concerns.

Referee #3:

[...]Most of my comments relate to minor ambiguities or missing citations. However, I do not have the impression that the RNase 6 dependence of *E. faecalis* detection is sufficiently supported by the data, so I would like to ask for further experiments and clarifications here.

I would like to request the following revisions:

Abstract:

"Herein, we characterized molecular RNase 6 cleavage mechanisms. BLaER1 RNASE6-/- cells showed a dampened TLR8-dependent response upon stimulation with isolated bacterial RNA (bRNA) but also upon infection with live whole bacteria."

Perhaps write "and" instead of "but"? I first wondered where the contradiction should be.

Thanks for kind suggestion, corrected as suggested: "Herein, we study RNase 6 for its role in TLR8 activation. BLaER1 cells, transdifferentiated into monocyte-like cells, as well as primary monocytes deficient for *RNASE6* show a dampened TLR8-dependent response upon stimulation with isolated bacterial RNA (bRNA) and also upon infection with live whole bacteria." (p. 2, l. 6)

Introduction:

Please add citations after the following sentences:

- Small synthetic agonists, such as the imidazoquinolinone derivate R848 and the benzodiazepine compound TL8-506, can efficiently activate the receptor by solely binding to the apex site.

We revised as suggested: "Small synthetic agonists, such as the imidazoquinolinone derivatives R848 and CL097 can efficiently activate the receptor solely by binding to the apical site (Tanji *et al*, 2013)." (p. 3, l. 13)

- Regarding their cleavage selectivity, it was shown that RNase T2 preferentially cleaves in between purines and uridine, and RNase A family members in general selectively cleave in between pyrimidines and purines.

Following the suggestions, we have reworded the text to "Regarding their cleavage selectivity, it was shown that RNase T2 preferentially cleaves between purines and uridine (Greulich *et al*, 2019), and RNase A family members in general cleave in between pyrimidines and purines (reviewed in (Boix *et al*, 2013))." (p. 3, l. 24)

- Among the several RNase A family members, RNase 2 was the first one to be shown to process ssRNA in the endosome.

We revised the text: "Among the several canonical RNase A family members, RNase 2 was the first one reported to be involved in the generation of ssRNA fragments in the endolysosome of THP-1 cells (Ostendorf *et al*, 2020)" (p. 3, l. 29)

- RNase 2 expression is mostly upregulated in eosinophils and basophils while in monocytes low expression has been observed.

We changed the respective sentence "Also known as Eosinophil-derived Neurotoxin (EDN), RNase 2 is mostly expressed in eosinophils, yet showing also expression in monocytes to lesser extent (Rosenberg, 2015). Thus, although ssRNA processing by RNase 2 may result in uridine-terminated breakdown products activating TLR8, its role in monocytes remains unsure." (p. 3, l. 31)

Please clarify this point:

"RNase 6 exhibits extracellular antimicrobial peptide (AMP) activity against a range of bacteria including *Escherichia coli* and *Staphylococcus aureus* by the induction of bacterial membrane depolarization and bacterial agglutination at low micromolar concentration (Pulido *et al.*, 2016).

Interestingly, RNase 6 upregulation upon urinary tract infection (UTI) with *E. coli* has been reported and RNASE6 transgenic mice showed increased protection upon experimental UTI with uropathogenic *E. coli* (Becknell et al., 2015; Ruiz-Rosado et al., 2023)."

Could you please interlink this part a bit better? When reading it, it is not directly clear why. Also, it seems contradictory to the first headline in the results part; please discuss / clarify this.

We reworked this text section to increase clarity: "RNase 6 exhibits extracellular antimicrobial peptide (AMP) activity against a range of bacteria at low micromolar concentrations (Becknell *et al*, 2015; Pulido *et al*, 2016). RNase 6 is highly cationic which might foster the binding to negatively charged components of the bacterial membrane thus inducing bacterial cell death (Becknell *et al*, 2015; Rosenberg & Dyer, 1996; Tong *et al*, 2023). Expression of RNase 6 seems to be limited to immune cells such as monocytes, neutrophils, and dendritic cells (Rosenberg & Dyer, 1996; Tong *et al*, 2023). It is detected at high concentrations in immune cell rich tissues such as the spleen and thymus (Becknell *et al*, 2015). RNase 6 was shown to protect against urinary tract infections (Becknell *et al*, 2015) and RNASE6 transgenic mice showed increased protection against infection with uropathogenic *E. coli* (Ruiz-Rosado *et al*, 2023). Of note, RNase 6 expression overlaps with TLR8 in monocytes, neutrophils, and myeloid DCs, and its role in the processing of ssRNA was recently reported (Tong *et al*, 2023). Yet, whether and how the ribonuclease activity of RNase 6 may contribute to the defense of bacterial infections remains unclear." (p. 4, l. 5).

We also noticed the controversial result from Figure 1C to a published paper, and we would like to discuss the following points.

1. From Becknell's paper (Becknell *et al*, 2015), the upregulation of RNase 6 was detected only upon infection with uropathogenic bacteria. During steady-state, both mRNA and protein level of RNase 6 were barely detectable in urinary tract and kidney from mice and humans. Upon UTI, lots of neutrophils and monocytes are recruited to lower urinary tract and kidney, which are known as major RNase 6 expressing cells, suggesting the positive correlation between upregulated RNase 6 level and infiltrating leukocytes.
2. As shown in Figure 1C, we infected RNase 6-sufficient transdifferentiated BLaER1 cells (Figure 2B and EV 2A) with bacteria for 1 hour and then stopped further infection by adding 50 µg/ml of Gentamycin. After another 2 hours incubation, cells were harvested and subjected to qPCR for RNASE6 expression. Our result demonstrates a reduced RNASE6 gene expression upon live bacterial infection, which may not be a direct comparison to published papers. Of note, Christensen-Quick et al (Christensen-Quick *et al*, 2016), reported a correlation between diminished RNase 6 in Th17 cells and inhibition of HIV replication, suggesting some pathogens may downregulate RNase 6 as immune evasion mechanism. Of course, further investigation will be required and this point is a bit out of our scope for this manuscript.
3. Interestingly and surprisingly, Becknell's data (Becknell *et al*, 2015) did not detect RNase 6 from human primary CD14⁺ cells in their publication, which we do see as abundant protein from the same cell population (Figure 2F and EV 2A).

Results and discussion:

"We measured proinflammatory cytokines in the cell-free supernatant of BLaER1 cells lacking RNASE6 upon infection with bacteria."

Could you rearrange this sentence? I know what you want to say, but at first reading, it sounds like these cells lack RNase 6 when they are infected.

We revised the sentence to avoid misunderstandings: “We infected cells with these two bacteria to assess the physiological role of RNase 6 and measured proinflammatory cytokines in the cell-free supernatant from BLaER1 cells lacking *RNASE6*.” (p. 7, l. 7)

Figure 2c and 2E:

Is *E. faecalis* sensing really depending on RNase 6 digestion? From these pictures, it does not look like this to me. The decrease in RNase 6 ko cells is really minimal, and in Figure 2E, IL-6 induction is even stronger in RNase 6 deficient cells. To keep these data in the paper, it would be essential to clarify this point, perhaps by RNase T2 ko and RNase 6 and T2 double ko.

Indeed, the RNase 6 dependency is weaker on live bacterial *E. faecalis* than *S. aureus* sensing. Yet, differences were significant (although small) and reproducible. To avoid overselling we reworded the text: “Live whole bacterial infection with *S. aureus* in *RNASE6*^{-/-} cells showed a strong and significant reduction of IL-6 and TNF- α secretion (Figure 2C, EV 2D). Despite the strong dependency on TLR8 recognition (Figure 1B), live infection with *E. faecalis* in RNase 6 deficient cells only resulted in a partial, yet significant reduction of cytokine secretion at highest MOI.” (p. 7, l. 10)

To further support our findings, we additionally performed gene editing of *RNASE6* in primary human CD14⁺ monocytes (hCD14⁺) via CRISPR-Cas9. The new data from hCD14⁺ are depicted in Figure 2F-I and EV 2H. As shown in Figure 2G, sensing of *S. aureus* bRNA but also of *E. faecalis* bRNA was clearly dependent on RNase 6 in primary human monocytes.

Figure 2D:

The two blots are impressively similar, so I checked several times to make sure that the same blot was not accidentally shown twice with different exposures. However, due to differences in the ladder intensity, I am now convinced that two different blots have been shown and that the work was just impressively accurate. Nevertheless, to be on the safe side, I would like to ask for this to be double-checked in-house.

We now show the whole original gel as Figure EV 2E. On the left side *E. faecalis* bRNA was analyzed, on the right side *S. aureus*. In the first manuscript the whole gel image was cut and shown as two subpanels. Due to restructuring of the Figure, we now show the whole gel image at once and thus clarify this point.

Figure 2E: Please discuss this figure in the results / discussion section.

Most important point: Please clarify RNase 6 and / or RNase T2 dependency of *E. faecalis* and discuss this difference in more detail.

We thank your comments. Although recognition of *E. faecalis* bRNA was reported before (Nishibayashi *et al*, 2015), the contribution of RNases in bRNA sensing has not been previously assessed. As shown in Figure 2D and 2G, we observe that proinflammatory cytokine secretion was strongly affected by the loss of RNase 6 in 2 different cell types (BLaER1 cells and, new data from primary monocytes). Of note, the effect of RNase 6 on recognizing *E. faecalis* bRNA is more pronounced in hCD14⁺ cells, though the expression of RNase T2 is not altered, suggesting RNase 6 plays an important role on sensing pathogen-derived ssRNA. To our knowledge, we are the first group demonstrating that RNase 6 is required in physiological setting. Based on our *ex cellulo* digestion of chimeric oligos (Figure 3 and EV 3), we propose that RNase 6 and RNase T2 contribute to TLR8 activation by generating uridine- or guanosine-terminated fragments, respectively.

We reworded the description of the subpanel:

“Although the dependence of *E. faecalis* RNA on TLR8 sensing has previously been shown (Nishibayashi *et al*, 2015), RNase dependence on this matter has not been previously assessed. Lack of RNase 6 affected the IL-6 response upon stimulation at a lower RNA concentration, but this effect was not sustained at a higher concentration of bRNA. We hypothesize this phenotype might be due to the effect of the remaining activity of other expressed RNases in the system, bRNA sequence composition or even

RNA modification which might limit RNase activity.” (p. 7, l. 27) As we are currently lacking data with respect to relative contributions of different RNase to overall sensing we cannot argue in more detail (see comment 2. on p. 3 to reviewer#1).

Additional References:

Becknell B, Eichler TE, Beceiro S, Li B, Easterling RS, Carpenter AR, James CL, McHugh KM, Hains DS, Partida-Sanchez S, *et al* (2015) Ribonucleases 6 and 7 have antimicrobial function in the human and murine urinary tract. *Kidney International* 87: 151–161

Bergstrøm B, Aune MH, Awuh JA, Kojen JF, Blix KJ, Ryan L, Flo TH, Mollnes TE, Espevik T & Stenvik J (2015) TLR8 Senses Staphylococcus aureus RNA in Human Primary Monocytes and Macrophages and Induces IFN- β Production via a TAK1–IKK β –IRF5 Signaling Pathway. *J Immunol* 195: 1100–1111

Boix E, Blanco JA, Nogués MV & Moussaoui M (2013) Nucleotide binding architecture for secreted cytotoxic endoribonucleases. *Biochimie* 95: 1087–1097

Christensen-Quick A, Lafferty M, Sun L, Marchionni L, DeVico A & Garzino-Demo A (2016) Human T h17 Cells Lack HIV-Inhibitory RNases and Are Highly Permissive to Productive HIV Infection. *Journal of Virology* 90: 7833–7847

Greulich W, Wagner M, Gaidt MM, Stafford C, Cheng Y, Linder A, Carell T & Hornung V (2019) TLR8 Is a Sensor of RNase T2 Degradation Products. *Cell* 179: 1264-1275.e13

Heinz LX, Lee J, Kapoor U, Kartnig F, Sedlyarov V, Papakostas K, César-Razquin A, Essletzbichler P, Goldmann U, Stefanovic A, *et al* (2020) TASL is the SLC15A4-associated adaptor for IRF5 activation by TLR7–9. *Nature* 581: 316–322

Livak KJ & Schmittgen TD (2001) Analysis of Relative Gene Expression Data Using Real-Time Quantitative PCR and the $2^{-\Delta\Delta CT}$ Method. *Methods* 25: 402–408

Nishibayashi R, Inoue R, Harada Y, Watanabe T, Makioka Y & Ushida K (2015) RNA of Enterococcus faecalis Strain EC-12 Is a Major Component Inducing Interleukin-12 Production from Human Monocytic Cells. *PLoS ONE* 10: e0129806

Ostendorf T, Zillinger T, Andryka K, Schlee-Guimaraes TM, Schmitz S, Marx S, Bayrak K, Linke R, Salgert S, Wegner J, *et al* (2020) Immune Sensing of Synthetic, Bacterial, and Protozoan RNA by Toll-like Receptor 8 Requires Coordinated Processing by RNase T2 and RNase 2. *Immunity* 52: 591-605.e6

Prats-Ejarque G, Arranz-Trullén J, Blanco JA, Pulido D, Nogués MV, Moussaoui M & Boix E (2016) The first crystal structure of human RNase 6 reveals a novel substrate-binding and cleavage site arrangement. *Biochem J* 473: 1523–1536

Pulido D, Arranz-Trullén J, Prats-Ejarque G, Velázquez D, Torrent M, Moussaoui M & Boix E (2016) Insights into the Antimicrobial Mechanism of Action of Human RNase6: Structural Determinants for Bacterial Cell Agglutination and Membrane Permeation. *Int J Mol Sci* 17: 552

Rademacher F, Dreyer S, Kopfnagel V, Gläser R, Werfel T & Harder J (2019) The Antimicrobial and Immunomodulatory Function of RNase 7 in Skin. *Front Immunol* 10: 2553

Rosenberg HF (2015) Eosinophil-Derived Neurotoxin (EDN/RNase 2) and the Mouse Eosinophil-Associated RNases (mEars): Expanding Roles in Promoting Host Defense. *Int J Mol Sci* 16: 15442–15455

Rosenberg HF & Dyer KD (1996) Molecular Cloning and Characterization of a Novel Human Ribonuclease (RNase k6): Increasing Diversity in the Enlarging Ribonuclease Gene Family. *Nucleic Acids Res* 24: 3507–3513

Ruiz-Rosado J de D, Cortado H, Kerckmar M, Li B, Ballash G, Cotzomi-Ortega I, Sanchez-Zamora YI, Gupta S, Ching C, Boix E, *et al* (2023) Human Ribonuclease 6 Has a Protective Role during Experimental Urinary Tract Infection. *J Innate Immun* 15: 865–875

Tanji H, Ohto U, Shibata T, Miyake K & Shimizu T (2013) Structural Reorganization of the Toll-Like Receptor 8 Dimer Induced by Agonistic Ligands. *Science* 339: 1426–1429

Tong A-J, Leylek R, Herzner A-M, Rigas D, Wichner S, Blanchette C, Tahtinen S, Kemball CC, Mellman I, Haley B, *et al* (2023) Nucleotide modifications enable rational design of TLR7-selective ligands by blocking RNase cleavage. *J Exp Med* 221: e20230341

Dear Prof. Dalpke,

Thank you for the submission of your revised manuscript. I have taken over its handling as my colleague Achim is currently not in the office. We have now received the enclosed reports from the referees and I am happy to say that all support its publication now. Only a few editorial requests will need to be addressed before we can proceed with the official acceptance of your manuscript.

- The Appendix file needs to be in portrait orientation; it also needs a title page with ms title and a table of content with page numbers; the nomenclature should be Appendix Table S1-S2 throughout the Appendix and ms file. Please correct.
- The Instructions and Example from the last page need to be removed from the Reagents and Tools table.
- The source data (SD) checklist should be uploaded as an individual "Related Manuscript File" and the SD files need to be saved as one folder per ms figure and then uploaded as .zip files.
- The synopsis image at the final image size of 550x600 pixels has some rather small text and the chemical structures are also very small. It would be better if you could send us a slightly more compact image with larger text. Thank you.
- Please remember that the PXD053881 dataset needs to be freely accessible upon the online publication of your ms, and that a specific URL for the PXD053881 dataset needs to be provided in the data availability section.
- Please note that the exact p values are not provided in the legends of figures 1b-c; 2a, c-e, g, i; 4c; EV 1a; EV 2d, f-h. Please add.
- Please indicate the statistical test used for data analysis in the legend of figure EV 2c.

I would like to suggest a few minor changes to the abstract. Please let me know whether you agree with the following:

TLR8 senses single-stranded RNA (ssRNA) fragments, processed via cleavage by ribonucleases (RNase) T2 and RNase A family members. Processing by these RNases releases uridines and purine-terminated residues resulting in TLR8 activation. Monocytes show high expression of RNase 6, yet this RNase has not been analyzed for its physiological contribution to the recognition of bacterial RNA by TLR8. Here, we show a role for RNase 6 in TLR8 activation. BLaER1 cells, transdifferentiated into monocyte-like cells, as well as primary monocytes deficient for RNASE6 show a dampened TLR8-dependent response upon stimulation with isolated bacterial RNA (bRNA) and also upon infection with live bacteria. Pretreatment of bacterial RNA with recombinant RNase 6 generates fragments that induce TLR8 stimulation in RNase 6 knockout cells. 2'O-RNA methyl modification, when introduced at the first uridine in the UA dinucleotide, impairs processing by RNase 6 and dampens TLR8 stimulation. In summary, our data show that RNase 6 processes bacterial RNA and generates uridine-terminated breakdown products that activate TLR8.

Referee #1:

The authors have adequately revised the manuscript text and figures according to my suggestions. They also done new experiments with primary monocytes, which support their main claim based on the cell model. These changes have improved the manuscript, and I have no further questions or concerns regarding publishing.

Referee #2:

The authors nicely improved the manuscript according to the reviewers suggestions. I would highly recommend publication of

the manuscript.

Referee #3:

The authors addressed all my concerns.

All editorial and formatting issues were resolved by the authors.

Prof. Alexander Dalpke
Heidelberg University
Dept. of Infectious Diseases, Medical Microbiology
Im Neuenheimer Feld 324
Baden-Wuerttemberg 69120
Germany

Dear Prof. Dalpke,

I am very pleased to accept your manuscript for publication in the next available issue of EMBO reports. Thank you for your contribution to our journal.

Yours sincerely,
